# *Tailoring*: ENCODING INDUCTIVE BIASES BY OPTIMIZING UNSUPERVISED OBJECTIVES AT PREDICTION TIME

## ABSTRACT

From CNNs to attention mechanisms, encoding inductive biases into neural networks has been a fruitful source of improvement in machine learning. Auxiliary losses are a general way of encoding biases in order to help networks learn better representations by adding extra terms to the loss function. However, since they are minimized on the training data, they suffer from the same generalization gap as regular task losses. Moreover, by changing the loss function, the network is optimizing a different objective than the one we care about. In this work we solve both problems: first, we take inspiration from *transductive learning* and note that, after receiving an input but before making a prediction, we can fine-tune our models on any unsupervised objective. We call this process tailoring, because we customize the model to each input. Second, we formulate a nested optimization (similar to those in meta-learning) and train our models to perform well on the task loss after adapting to the tailoring loss. The advantages of tailoring and meta-tailoring are discussed theoretically and demonstrated empirically on several diverse examples: encoding inductive conservation laws from physics, increasing robustness to adversarial examples, meta-tailoring with contrastive losses to improve theoretical generalization guarantees, and increasing performance in model-based RL.

## 1 INTRODUCTION

The key to successful generalization in machine learning is the encoding of useful inductive biases. A variety of mechanisms, from parameter tying to data augmentation, have proven useful but there is no systematic strategy for designing and implementing these biases.

Auxiliary losses are a paradigm for encoding a wide variety of biases, constraints and objectives, helping networks learn better representations and generalize more broadly. They add an extra term to the task loss and minimize it over the training data or, in semi-supervised learning, on an extra set of unlabeled data. However, they have two major difficulties:

1. Auxiliary losses are only minimized at training time, but not for the query points. This causes a generalization gap between training and testing, in addition to that of the task loss.

2. By minimizing the sum of the task loss plus the auxiliary loss, we are optimizing a different objective than the one we care about (only the task loss).

In this work we propose a solution to each problem:

1. We use ideas from *transductive learning* to minimize the auxiliary loss at the query by running an optimization at prediction time, eliminating the generalization gap for the auxiliary loss. We call this process *tailoring*, because we customize the model to each query.

2. We use ideas from *meta-learning* to learn a model that performs well on the task loss assuming that we will be optimizing the auxiliary loss. This *meta-tailoring* effectively trains the model to leverage the unsupervised tailoring loss to minimize the task loss.

**Tailoring a predictor** In classical inductive supervised learning, an algorithm consumes a training dataset of input-output pairs, $((x_i, y_i))_{i=1}^n$, and produces a set of parameters $\hat{\theta}$ by minimizing a supervised loss $\sum_{i=1}^n \mathcal{L}^{\text{sup}}(f_\theta(x_i), y_i)$ and, optionally, an unsupervised auxiliary loss $\sum_{i=1}^n \mathcal{L}^{\text{unsup}}(\theta, x_i)$. These parameters specify a hypothesis $f_{\hat{\theta}}(\cdot)$ that, given a new input $x$, generates an output $\hat{y} = f_{\hat{\theta}}(x)$. This problem setting misses a substantial opportunity: before the learning algorithm sees the query

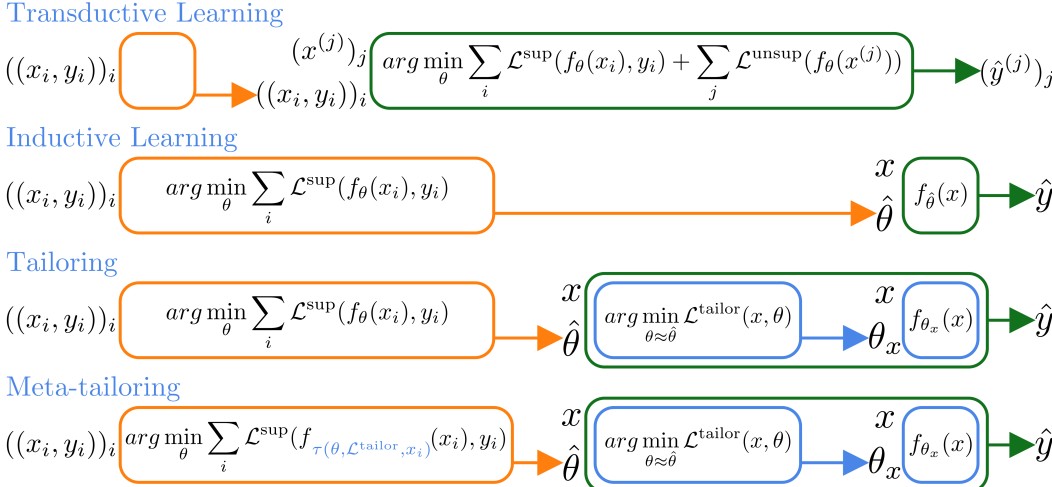

Figure 1: Comparison of several learning settings with *offline* computation in the orange boxes and *online* computation in the green boxes, with tailoring in blue. For meta-tailoring training, $\tau(\theta, \mathcal{L}^{\text{tailor}}, x) = \arg\min_{\theta' \approx \theta} \mathcal{L}^{\text{tailor}}(x, \theta')$ represents the tailoring process resulting in $\theta_x$. Although tailoring and meta-tailoring are best understood in supervised learning, they can also be applied in reinforcement learning, as shown in section 5.3.

point $x$, it has distilled the data down to a set of parameters, which are frozen during inference, and so it cannot use new information about the *particular $x$* that it will be asked to make a prediction for.

Vapnik recognized an opportunity to make more accurate predictions when the query point is known, in a framework that is now known as *transductive learning* (Vapnik, 1995; Chapelle et al., 2000). In transductive learning, a single algorithm consumes both labeled data, $((x_i, y_i))_{i=1}^n$, and a set of input points for which predictions are desired, $(x^{(j)})_j$, and produces predicted outputs $(\hat{y}^{(j)})_j$ for each of the queries, as illustrated in the top row of figure 1. In general, however, we do not know queries *a priori*, and instead we want an inductive rule that makes predictions on-line, as queries arrive. To obtain a prediction function from a transductive system, we would need to encapsulate the entire learning procedure inside the prediction function. This strategy would achieve our objective of taking $x$ into account at prediction time, but would be computationally much too slow.

We observe that this strategy for combining induction and transduction would perform very similar computations for each prediction, sharing the same training data and objective. We can use ideas from meta-learning to find a shared "meta-hypothesis" that can then be efficiently adapted to each query (treating each prediction as a task). As shown in the third row of figure 1, we first run regular supervised learning to obtain parameters $\hat{\theta}$; then, given a query input $x$, we fine-tune $\hat{\theta}$ on an unsupervised loss ($\mathcal{L}^{\text{tailor}}$) to obtain customized parameters $\theta_x$ and use them to make the final prediction: $f_{\theta_x}(x)$. We call this process *tailoring*, because we adapt the model to each particular input for a customized fit. Notice that tailoring optimizes the loss at the query point, eliminating the generalization gap on the auxiliary loss.

**Meta-tailoring** Since we will be applying tailoring at prediction time, it is natural to anticipate this adaptation during training, resulting in a two-layer optimization similar to those used for meta-learning. Because of this similarity, we call this process, illustrated in the bottom row of figure 1, *meta-tailoring*. Now, rather than letting $\hat{\theta}$ be the direct minimizer of the supervised loss, we set it to

$$\hat{\theta} \in \arg\min_\theta \sum_{i=1}^n \mathcal{L}^{\text{sup}}(f_{\tau(\theta, \mathcal{L}^{\text{tailor}}, x_i)}(x_i), y_i).$$

Notice that by optimizing this nested objective, the outer process is now optimizing the only objective we care about, $\mathcal{L}^{\text{sup}}$, instead of a proxy combination of $\mathcal{L}^{\text{sup}}$ and $\mathcal{L}^{\text{unsup}}$. At the same time, we are learning to leverage the unsupervised tailoring losses in the inner optimization to affect the model before making the final prediction, both during training and at prediction time.

In many settings, we wish to make predictions for a large number of query points in a (mini-)batch, but because tailoring adapts to every *point*, we must take extra care to be sure it can run efficiently in parallel. Inspired by conditional normalization (Dumoulin et al., 2016) we propose adding element-wise affine transformations and only adapting these parameters in the inner optimization. This allows us to *tailor* outputs for multiple inputs in parallel, without inputs affecting each other's computations. We prove theoretically, in section 4, and provide experimental evidence, in section 5.1, that optimizing these parameters alone has enough capacity to minimize a large class of tailoring losses.

**Losses for tailoring**    Tailoring can be used any time we are making a prediction and have an unsupervised loss that can be minimized on the current input; although we mostly focus on supervised learning, tailoring can also be applied to reinforcement learning, as shown in section 5.3. There are many types of *tailoring* losses that may be useful; here we enumerate three broad classes.

 Losses imposing priors and constraints satisfied by the correct predictions, such as conservation of momentum and energy when learning a physics model, symmetry under specific transformations, or cycle-consistency when learning to translate between two languages. We show that tailoring a loss of this type improves predictive losses in section 5.1, where we model a planetary system. We can also leverage soft priors; for instance, in section 5.3 we show how encouraging action-conditional predictions to be likely under an action-unconditional model can improve model-based RL.

 Losses that help learn better representations, such as the contrastive losses Hadsell et al. (2006) explored in semi-supervised learning, or learning to predict one part of the input from another part, such as depth from RGB (Mirowski et al., 2016). In section 3.1 we show how tailoring with a contrastive loss improves supervised prediction performance.

 Losses informed by theoretical guarantees. The guarantees of many theorems depend on unsupervised quantities, such as smoothness or distance to the prediction boundary. By optimizing such quantities on the query, or on the surrounding area, we can get better guaranteed performance. In section 5.2, we show this on adversarial examples, where smooth predictions around the test point are critical.

**Contributions**    In summary, our contributions are the following:

1. Introducing *tailoring*, a new framework for encoding inductive biases by minimizing unsupervised losses at prediction time, with theoretical guarantees and broad potential applications.

2. Formulating *meta-tailoring*, which adjusts the outer objective to optimize only the task loss, and developing a new algorithm, CNGRAD, for efficient meta-tailoring.

3. Demonstrating *tailoring* in four domains: encoding conservation laws in a physics prediction problem, increasing resistance to adversarial examples by increasing local smoothness at prediction time, making model-based reinforcement learning more data-efficient, and improving theoretical guarantees of prediction quality by tailoring with a contrastive loss.

## 2    RELATED WORK

There are other learning frameworks that perform optimization at prediction time, such as energy-based models (Ackley et al., 1985; Hinton, 2002; LeCun et al., 2006) or models that embed optimization layers in neural networks, whose outputs are the solution of an optimization problem defined by the previous layer (Amos & Kolter, 2017; Tschiatschek et al., 2018). In contrast to these lines of work, we optimize the parameters of the model, not the hidden activations or the output.

Meta-learning (Schmidhuber, 1987; Bengio et al., 1995; Thrun & Pratt, 1998) has the same two-level optimization structure as our work, but focuses on multiple prediction tasks, each of which has its own separate training data. Most optimization-based meta-learning algorithms can be converted to the meta-tailoring setting. There is a particularly clear connection to MAML (Finn et al., 2017), when we let the tailoring method be a step of gradient descent: $\tau(\hat{\theta}, \mathcal{L}^{\text{tailor}}, x) = \hat{\theta} - \lambda \nabla_{\hat{\theta}} \mathcal{L}^{\text{tailor}}(x, \hat{\theta})$. There are other optimization-based approaches to meta-learning whose adaptations can be batched (Zintgraf et al., 2018; Rakelly et al., 2019; Alet et al., 2019). In particular, FiLM networks (Perez et al., 2018), which predict customized conditional normalization layers, have been used in meta-learning (Zintgraf et al., 2018; Requeima et al., 2019). By optimizing the conditional normalization layers themselves, our method CNGRAD is simpler, while remaining provably sufficiently expressive. More importantly, we can add to a trained model CNGRAD layers with weights initialized to the identity and adapt them to perform tailoring or fine-tune the model with a meta-tailoring objective.

Tailoring is inspired by transductive learning. However, transductive methods, because they operate on a batch of unlabeled points, are able to make use of the underlying distributional properties of those points. On the other hand, tailoring does not need to receive the queries before doing the bulk of the computation. Within transductive learning, local learning (Bottou & Vapnik, 1992) has input-dependent parameters, but it uses similarity in raw input space to select a few data-points instead of reusing the prior learned across the whole data. Some methods (Garcia & Bruna, 2017; Liu et al., 2018) in meta-learning propagate predictions along the test samples in a classic transductive fashion.

Optimization processes similar to tailoring and meta-tailoring have been proposed before, to adapt to different types of variations between training and testing. Sun et al. (2019) propose to adapt to a change of distribution with few samples by unsupervised fine-tuning at test-time, applying it with a loss of predicting whether the input has been rotated. Other methods in the meta-learning setting exploit test samples of a new task by minimizing either entropy (Dhillon et al., 2020) or a learned loss (Antoniou & Storkey, 2019) in the inner optimization. Finally, Wang et al. (2019) uses entropy in the inner optimization to adapt to large scale variations in image segmentation. In contrast, we propose (meta-)tailoring as a general effective way to impose inductive biases in the classic machine learning setting. We also unify tailoring and meta-tailoring with arbitrary losses under one paradigm, theoretically and empirically showing the advantage of each modification to classic inductive learning. Appendix F shows experimental results analyzing why using prior work on adapting to the test distribution performs worse than tailoring (which, in turn, performs worse than meta-tailoring) in the classic ML setting where test and training come from the same distribution.

## 3 THEORETICAL MOTIVATIONS OF META-TAILORING

In this section, we study potential advantages of meta-tailoring from the theoretical viewpoint, formalizing the intuitions conveyed in the introduction. By acting symmetrically during training and prediction time, meta-tailoring allows us to closely relate its training and expected losses, whereas tailoring in general may make them less related.

### 3.1 META-TAILORING WITH A CONTRASTIVE TAILORING LOSS

*Contrastive learning* (Hadsell et al., 2006) has seen significant successes recently in problems of semi-supervised learning from images (Oord et al., 2018; He et al., 2019; Chen et al., 2020). The main idea is to create multiple versions of each training image, and learn a representation in which variations of the same image are very close and variations of different images are far apart. Typical variations involve cropping, color distortions and rotation. We show theoretically that, under reasonable conditions, meta-tailoring using a particular contrastive loss $\mathcal{L}_{\text{cont}}$ as $\mathcal{L}^{\text{tailor}} = \mathcal{L}_{\text{cont}}$ helps us improve generalization errors in expectation compared with performing classical inductive learning.

When using meta-tailoring, we define $\theta_{x,S}$ to be the $\theta_x$ obtained with a training dataset $S = ((x_i, y_i))_{i=1}^n$ and tailoring with the contrastive loss at the prediction point $x$. Theorem 1 provides an upper bound on the expected supervised loss $\mathbb{E}_{x,y}[\mathcal{L}^{\text{sup}}(f_{\theta_{x,S}}(x), y)]$ in terms of the expected contrastive loss $\mathbb{E}_x[\mathcal{L}_{\text{cont}}(x, \theta_{x,S})]$ (which is defined and analyzed in more detail in Appendix B), the empirical supervised loss $\frac{1}{n}\sum_{i=1}^n \mathcal{L}^{\text{sup}}(f_{\theta_{x_i,S}}(x_i), y_i)$ of the meta-tailoring algorithm, and the uniform stability $\zeta$ of the meta-tailoring algorithm. As a complement, instead of using the uniform stability $\zeta$, Theorem 6 (detailed in appendix C) provides a similar bound with the Rademacher complexity (Bartlett & Mendelson, 2002) $\mathcal{R}_n(\mathcal{L}^{\text{sup}} \circ \mathcal{F})$ of the set $\mathcal{L}^{\text{sup}} \circ \mathcal{F} = \{(x, y) \mapsto \mathcal{L}^{\text{sup}}(f_{\theta_x}(x), y) : (x \mapsto f_{\theta_x}(x)) \in \mathcal{F}\}$. All proofs in this paper are deferred to Appendix C.

**Definition 1.** Let $S = ((x_i, y_i))_{i=1}^n$ and $S' = ((x'_i, y'_i))_{i=1}^n$ be any two training datasets that differ by a single point. Then, a meta-tailoring algorithm $S \mapsto f_{\theta_{\mathbf{x},S}}(x)$ is *uniformly $\zeta$-stable* if $\forall (x, y) \in \mathcal{X} \times \mathcal{Y}, \; |\mathcal{L}^{\text{sup}}(f_{\theta_{\mathbf{x},S}}(x), y) - \mathcal{L}^{\text{sup}}(f_{\theta_{\mathbf{x},S'}}(x), y)| \leq \frac{\zeta}{n}$.

**Theorem 1.** *Let $S \mapsto f_{\theta_{\mathbf{x},S}}(x)$ be a uniformly $\zeta$-stable meta-tailoring algorithm. Then, for any $\delta > 0$, with probability at least $1 - \delta$ over an i.i.d. draw of $n$ i.i.d. samples $S = ((x_i, y_i))_{i=1}^n$, the following holds: for any $\kappa \in [0, 1]$, $\mathbb{E}_{x,y}[\mathcal{L}^{\text{sup}}(f_{\theta_{\mathbf{x},S}}(x), y)] \leq \kappa \mathbb{E}_x\left[\mathcal{L}_{\text{cont}}(x, \theta_{\mathbf{x},S})\right] + (1 - \kappa)\mathcal{J}$, where $\mathcal{J} = \frac{1}{n}\sum_{i=1}^n \mathcal{L}^{\text{sup}}(f_{\theta_{x_i,S}}(x_i), y_i) + \frac{\zeta}{n} + (2\zeta + c)\sqrt{(\ln(1/\delta))/(2n)}$, and $c$ is the upper bound on the per-sample loss as $\mathcal{L}^{\text{sup}}(f_\theta(x), y) \leq c$.*

In the case of regular inductive learning we get a bound of the exact same form, except that we have a single $\theta$ instead of a $\theta_{\mathbf{x}}$ tailored to each input $x$. These differences are marked in bold

green in Definition 1 and Theorem 1. This theorem illustrates the effect of meta-tailoring on contrastive learning, with its potential reduction of the expected contrastive loss $\mathbb{E}_x[\mathcal{L}_{\text{cont}}(x, \theta_{x,S})]$. In classic induction, we may aim to minimize the empirical contrastive loss $\frac{1}{\bar{n}} \sum_{i=1}^{\bar{n}} \mathcal{L}_{\text{cont}}(x_i, \theta)$ with $\bar{n}$ potentially unlabeled training samples (in addition to the empirical supervised loss), which incurs the additional generalization error of $\mathbb{E}_x[\mathcal{L}_{\text{cont}}(x, \theta_{x,S})] - \frac{1}{\bar{n}} \sum_{i=1}^{\bar{n}} \mathcal{L}_{\text{cont}}(x_i, \theta)$. In contrast, the meta-tailoring can avoid this extra generalization error by directly minimizing $\mathbb{E}_x[\mathcal{L}_{\text{cont}}(x, \theta_{x,S})]$.

In the case where $\mathbb{E}_x[\mathcal{L}_{\text{cont}}(x, \theta_{x,S})]$ is left large (e.g., due to large computational cost at prediction time), Theorem 1 still illustrates competitive generalization bounds of meta-tailoring with small $\kappa$, when compared to classical induction. For example, with $\kappa = 0$, it provides standard types of generalization bounds with the uniform stability for meta-tailoring algorithms. Even in this case, the bounds are not equivalent to those of classic induction, and there are potential benefits of meta-tailoring, which are discussed in the following section with a more general setting.

### 3.2 META-TAILORING WITH GENERAL TAILORING LOSSES

The benefits of meta-tailoring go beyond contrastive learning: the following remark provides the generalization bounds for meta-tailoring with any tailoring loss $\mathcal{L}^{\text{tailor}}(x, \theta)$ and an arbitrary supervised loss $\mathcal{L}^{\text{sup}}(f_\theta(x), y)$.

**Remark 1.** *For any function $\varphi$ such that $\mathbb{E}_{x,y}[\mathcal{L}^{\text{sup}}(f_\theta(x), y)] \leq \mathbb{E}_x[\varphi(\mathcal{L}^{\text{tailor}}(x, \theta))]$, Theorems 1 and 6 hold with the map $\mathcal{L}_{\text{cont}}$ being replaced by the function $\varphi \circ \mathcal{L}^{\text{tailor}}$.*

Remark 1 shows the benefits of meta-tailoring through its effects on three factors: the expected unlabeled loss $\mathbb{E}_x[\varphi(\mathcal{L}^{\text{tailor}}(x, \theta_{x,S}))]$, uniform stability $\zeta$, and Rademacher complexity $\mathcal{R}_n(\mathcal{L}^{\text{sup}} \circ \mathcal{F})$.

It is important to note that meta-tailoring can directly minimize the expected unlabeled loss $\mathbb{E}_x[\varphi(\mathcal{L}^{\text{tailor}}(x, \theta_{x,S}))]$, whereas classic induction can only minimize its empirical version, which results in the additional generalization error on the difference between the expected unlabeled loss and its empirical version. For example, if $\varphi$ is monotonically increasing and $\mathcal{L}^{\text{tailor}}(x, \theta)$ represents the physical constraints at each input $x$ (as in the application in section 5.1), then classic induction requires the physical constraints of neural networks at the *training* points to generalize to the physical constraints at *unseen* (e.g., testing) points. Meta-tailoring avoids this requirement by directly minimizing violations of the physical constraints at each point at prediction time.

Another potential benefit of meta-tailoring can be understood through the improvement in the *parameter stability* $\zeta_\theta$ defined such that $\forall (x,y) \in \mathcal{X} \times \mathcal{Y}, \|\theta_{x,S} - \theta_{x,S'}\| \leq \frac{\zeta_\theta}{n}$, for all $S, S'$ differing by a single point. In the case of the meta-tailoring method of $\theta_{x,S} = \hat{\theta}_S - \lambda \nabla \mathcal{L}^{\text{tailor}}(x, \hat{\theta}_S)$, we can obtain an improvement on the parameter stability $\zeta_\theta$ if $\nabla \mathcal{L}^{\text{tailor}}(x, \hat{\theta}_S)$ can pull $\hat{\theta}_S$ and $\hat{\theta}_{S'}$ closer so that $\|\theta_{x,S} - \theta_{x,S'}\| < \|\hat{\theta}_S - \hat{\theta}_{S'}\|$, which is ensured, for example, if $\| \cdot \| = \| \cdot \|_2$ and $\cos(v_1, v_2) \frac{\|v_1\|}{\|v_2\|} > \frac{1}{2}$ where $\cos(v_1, v_2)$ is the cosine similarity of $v_1$ and $v_2$, with $v_1 = \hat{\theta}_S - \hat{\theta}_{S'}$, $v_2 = \lambda(\nabla \mathcal{L}^{\text{tailor}}(x, \hat{\theta}_S) - \nabla \mathcal{L}^{\text{tailor}}(x, \hat{\theta}_{S'}))$ and $v_2 \neq 0$. Here, the uniform stability $\zeta$ and the parameter stability $\zeta_\theta$ are closely related as $\zeta \leq C\zeta_\theta$, where $C$ is the upper bound on the Lipschitz constants of the maps $\theta \mapsto \mathcal{L}^{\text{sup}}(f_\theta(x), y)$ over all $(x, y) \in \mathcal{X} \times \mathcal{Y}$ under the norm $\| \cdot \|$, since $|\mathcal{L}^{\text{sup}}(f_{\theta_{x,S}}(x), y) - \mathcal{L}^{\text{sup}}(f_{\theta_{x,S'}}(x), y)| \leq C\|\theta_{x,S} - \theta_{x,S'}\| \leq \frac{C\zeta_\theta}{n}$.

## 4 CNGRAD: A SIMPLE ALGORITHM FOR EXPRESSIVE, EFFICIENT TAILORING

In the previous sections, we have discussed the principal motivation and theoretical advantages of (meta-)tailoring. However, there is a remaining issue for efficient GPU computations: whereas one can efficiently parallelize the evaluation of a single model over a batch across inputs, it is challenging to efficiently parallelize the evaluation of multiple *tailored* models over a batch. To overcome this issue, by building on CAVIA (Zintgraf et al., 2018) and WarpGrad (Flennerhag et al., 2019), we propose CNGRAD which adapts only *conditional normalization* parameters and enables efficient GPU computations for (meta-)tailoring. CNGRAD can also be used in regular meta-learning; details and pseudo-codes of both versions can be found in appendix D.

As is done in batch-norm (Ioffe & Szegedy, 2015) after element-wise normalization, we can implement an element-wise affine transformation with parameters $(\gamma, \beta)$, scaling and shifting the output $h_k^{(l)}(x)$ of each $k$-th node at $l$-th hidden layer independently: $\gamma_k^{(l)} h_k^{(l)}(x) + \beta_k^{(l)}$. In conditional

normalization, Dumoulin et al. (2016) train a collection of $(\gamma, \beta)$ in a multi-task fashion to learn different tasks with a single network. We propose to bring this concept to the meta-learning and (meta-)tailoring settings and adapt the affine parameters $(\gamma, \beta)$ to each query . For meta-tailoring, the inner optimization minimizes the tailoring loss at an input $x$ by adjusting the affine parameters and the outer optimization adapts the rest of the network. Similar to MAML (Finn et al., 2017), we implement a first-order version, which does not backpropagate through the optimization, and a second-order version, which does. We can efficiently parallelize computations of multiple tailored models over a (mini-)batch in a GPU in the same way as that in a classic induction model, because the adapted parameters only require element-wise multiplications and additions.

CNGRAD is widely applicable, since we can add these adaptable affine parameters to any hidden layer.While we are only changing a tiny portion of the network, we prove below that, under realistic assumptions, we can minimize the inner tailoring loss using only the affine parameters. However, it is worth noting that we still need the entire network to minimize the outer meta-objective.

To analyze properties of the adaptable affine parameters, let us decompose $\theta$ into $\theta = (w, \gamma, \beta)$, where the $w$ contains all the weight parameters (including bias terms), and the $(\gamma, \beta)$ contains all the affine parameters. Given arbitrary coefficients $\eta_1, \ldots, \eta_{n_g} \in \mathbb{R}$ and an arbitrary function $(f_\theta(x), x) \mapsto \ell^{\text{tailor}}(f_\theta(x), x)$, let $\mathcal{L}^{\text{tailor}}(x, \theta) = \sum_{i=1}^{n_g} \eta_i \ell_{\text{tailor}}(f_\theta(g^{(i)}(x)), x)$, where $g^{(1)}, \ldots, g^{(n_g)}$ are arbitrary input augmentation functions at prediction time. Note that $n_g$ is typically small ($n_g \ll n$) in meta-tailoring; e.g., $n_g = 1$ in the method in Section 5.1 regardless of the size of the dataset $n$.

Corollary 1 states that for any given $\hat{w}$, if we add any non-degenerate Gaussian noise $\delta$ as $\hat{w} + \delta$ with zero mean and any variance on $\delta$, the global minimum value with respect to all parameters $(w, \gamma, \beta)$ can be achieved by optimizing only the affine parameters $(\gamma, \beta)$, with probability one.

**Assumption 1.** *(Common activation)* The activation function $\sigma(x)$ is real analytic, monotonically increasing, and the limits exist as: $\lim_{x \to -\infty} \sigma(x) = \sigma_- > -\infty$ and $\lim_{x \to +\infty} \sigma(x) = \sigma_+ \leq +\infty$.

**Theorem 2.** *For any $x \in \mathcal{X}$ that satisfies $\|g^{(i)}(x)\|_2^2 - g^{(i)}(x)^\top g^{(j)}(x) > 0$ (for all $i \neq j$), and for any fully-connected neural network with a single output unit, at least $n_g$ neurons per hidden layer, and activation functions that satisfy Assumption 1, the following holds: $\inf_{w, \gamma, \beta} \mathcal{L}^{\text{tailor}}(x, w, \gamma, \beta) = \inf_{\gamma, \beta} \mathcal{L}^{\text{tailor}}(x, \bar{w}, \gamma, \beta)$ for any $\bar{w} \notin \mathcal{W}$ where Lebesgue measure of $\mathcal{W} \subset \mathbb{R}^d$ is zero.*

**Corollary 1.** *Under the assumptions of Theorem 2, for any $\hat{w} \in \mathbb{R}^d$, with probability one over randomly sampled $\delta \in \mathbb{R}^d$ accordingly to any non-degenerate Gaussian distribution, the following holds: $\inf_{w, \gamma, \beta} \mathcal{L}^{\text{tailor}}(x, w, \gamma, \beta) = \inf_{\gamma, \beta} \mathcal{L}^{\text{tailor}}(x, \hat{w} + \delta, \gamma, \beta)$ for any $x \in \mathcal{X}$.*

The assumption and condition in theorem 2 are satisfied in practice(see Appendix A for more details). Therefore, CNGRAD is a practical and computationally efficient method to implement (meta-)tailoring.

## 5 EXPERIMENTS

### 5.1 TAILORING TO IMPOSE SYMMETRIES AND CONSTRAINTS AT PREDICTION TIME

Constructing inductive biases that exploit invariances and symmetries is an established strategy for boosting performance in machine learning. During training, we often regularize our networks to satisfy certain criteria; however, this does not guarantee that these criteria will be satisfied outside the training dataset (Suh & Tedrake, 2020). Another option is to construct *ad hoc* neural network architectures to exploit constraints for important problems, such as using Hamiltonian neural networks (Greydanus et al., 2019) to impose energy (but not momentum) conservation. Tailoring provides a general solution to this problem by adapting the model at prediction time to satisfy the criteria. In meta-tailoring, we also train the system to make good predictions after satisfying the constraint.

We demonstrate this use of tailoring by enforcing physical conservation laws to more accurately predict the evolution of a 5-body planetary system governed by gravitational forces. This prediction problem is challenging, as $m$-body systems become chaotic for $m > 2$. We generate a supervised-learning dataset with positions, velocities and masses of all 5 bodies as inputs and the changes in position and velocity at the next time-step as targets. Appendix E describes the dataset in greater detail.

To predict the data, we use a 3-layer feed-forward network and apply a tailoring loss based on the laws of conservation of energy and momentum. More concretely, we take the original predictions and

| Method | loss | relative |
|---|---|---|
| Inductive learning | .041 | - |
| Opt. output(50 st.) | .041 | $(0.7 \pm 0.1)\%$ |
| 6400-spl. TTT(50st.) | .040 | $(3.6 \pm 0.2)\%$ |
| Tailoring(1 step) | .040 | $(1.9 \pm 0.2)\%$ |
| Tailoring(5 st.) | .039 | $(6.3 \pm 0.3)\%$ |
| Tailoring(10 st.) | .038 | $(7.5 \pm 0.1)\%$ |
| Meta-tailoring(0 st.) | .030 | $(26.3 \pm 3.3)\%$ |
| Meta-tailoring(1 st.) | .029 | $(29.9 \pm 3.0)\%$ |
| Meta-tailoring(5 st.) | .027 | $(35.3 \pm 2.6)\%$ |
| Meta-tailoring(10 st.) | .026 | $(36.0 \pm 2.6)\%$ |

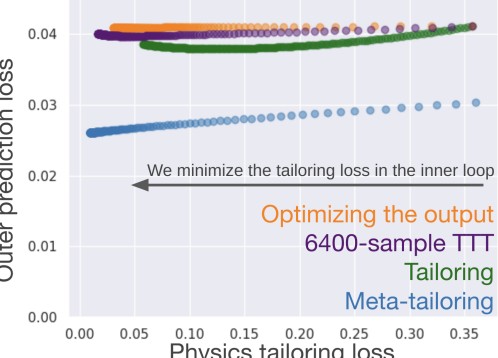

Table 1: Test MSE loss for different methods; the second column shows the relative improvement over basic inductive supervised learning. The test-time training (TTT) baseline uses a full batch of 6400 test samples to adapt, not allowed in regular SL. With a few gradient steps, tailoring significantly over-performs all baselines. Meta-tailoring improves even further, with $35\%$ improvement.

Figure 2: Optimization at prediction time on the planet data; each path going from right to left as we minimize the physics tailoring loss. We use a small step size to illustrate the path. Tailoring and the two baselines share only differ in their test-time computations, thus sharing their starts. Meta-tailoring has a lower starting loss, faster optimization, and no overfitting to the tailoring loss.

adapt our model using the $L_1$ loss between the initial and final energy and momentum of the whole system. Ensuring this conservation can improve predictions, but notice that minimizing the tailoring loss alone does not guarantee good predictions: predicting that the output equals the input conserves energy and momentum perfectly, but is not correct.

Tailoring adapts some parameters in the network in order to improve the tailoring loss. An alternative for enforcing conservation would be to adapt the output $y$ value directly. Table 1 compares the predictive accuracy of inductive learning to direct output optimization as well as tailoring and meta-tailoring, using varying numbers of gradient steps. We observe that tailoring is more effective than adapting the output, as the parameters provide a prior on what changes are more natural. For meta-tailoring, we try both first-order and second-order versions of CNGRAD: the first-order version gave slightly better results, possibly because it was trained with a higher tailor learning rate ($10^{-3}$) with which the second-order version meta-training was unstable (we thus used $10^{-4}$). More details can be found in Appendix E. Interestingly, meta-tailoring without any query-time tailoring steps already performs much better than the original model, even though both models have almost the same number of parameters and can overfit the dataset.We conjecture meta-tailoring training is adding an inductive bias that guides optimization towards learning a more generalizable model, even without tailoring at test time. Finally, plot 2 shows the prediction-time optimization paths for different methods.

## 5.2    TAILORING FOR ROBUSTNESS AGAINST ADVERSARIAL EXAMPLES

Despite their successes, neural networks remain susceptible to the problem of adversarial examples (Biggio et al., 2013; Szegedy et al., 2013): targeted small perturbations of an input can cause the network to mis-classify it. One approach is to make the prediction function smooth via adversarial training (Madry et al., 2017); however, this only ensures smoothness in the training points and constraining our model to be smooth everywhere makes it lose capacity. This is a perfect opportunity for applying (meta-)tailoring, since we can ask for smoothness *a posteriori*, only on the specific query.

We apply meta-tailoring to robustly classifying CIFAR-10 (Krizhevsky et al., 2009) and ImageNet (Deng et al., 2009) images, tailoring predictions so that they are locally smooth. We meta-tailor our classifier using (the first-order version of) CNGRAD and a tailoring loss that enforces smoothness on the entire vector of features of the penultimate layer in the neural network (denoted $g_\theta(x)$):

$$\mathcal{L}^{\text{tailor}}(x, \theta) = \mathbb{E}[\text{cos\_dist}(g_\theta(x), g_\theta(x + \delta))], \quad \delta \sim N(0, \nu^2), \tag{1}$$

where $\text{cos\_dist}(v_1, v_2)$ is the cosine distance between vectors $v_1$ and $v_2$. This loss is inspired by Ilyas et al. (2019), who argue that adversarial examples are caused by features which are predictive, but non-robust to perturbations. In that way, our loss adjusts the model to ensure features are locally

| $\sigma$ | Method | 0.0 | 0.5 | 1.0 | 1.5 | 2.0 | 2.5 | 3.0 | ACR |
|---|---|---|---|---|---|---|---|---|---|
| 0.25 | (Inductive) RandSmooth | 0.67 | 0.49 | 0.00 | 0.00 | 0.00 | 0.00 | 0.00 | 0.470 |
| | Meta-tailored | **0.72** | **0.55** | 0.00 | 0.00 | 0.00 | 0.00 | 0.00 | **0.494** |
| 0.50 | (Inductive) RandSmooth | 0.57 | 0.46 | 0.37 | 0.29 | 0.00 | 0.00 | 0.00 | 0.720 |
| | Meta-tailored | 0.66 | 0.54 | **0.42** | **0.31** | 0.00 | 0.00 | 0.00 | **0.819** |
| 1.00 | (Inductive) RandSmooth | 0.44 | 0.38 | 0.33 | 0.26 | 0.19 | 0.15 | 0.12 | 0.863 |
| | Meta-tailored | 0.52 | 0.45 | 0.36 | **0.31** | **0.24** | **0.20** | **0.15** | **1.032** |

Figure 3: Percentage of points with certificate above different radii, and average certified radius (ACR) for on the ImageNet dataset. Meta-tailoring improves the Average Certification Radius by $5.1\%, 13.8\%, 19.6\%$ respectively. Results for Cohen et al. (2019) are taken from Zhai et al. (2020).

robust at the input query before making a prediction. To keep inference fast, we approximate this loss with a single sample, but it could be improved with more samples or more sophisticated methods.

We build on the work of Cohen et al. (2019), who developed a method for certifying the robustness of a model via randomized smoothing. It samples several points from a Gaussian $N(x, \sigma^2)$ around the query and, if there is enough agreement in classification, it provides a certificate that the example cannot be adversarially modified by a small perturbation to have a different class. We show that meta-tailoring improves the original randomized smoothing method, testing for $\sigma = 0.25, 0.5, 1.0$. For simplicity, we use $\nu = 0.1$ for all experiments. We initialized with the weights of Cohen et al. (2019) to speed up training in all ImageNet experiments and to avoid training divergence for CIFAR-10, $\sigma = 1$ (this divergence had already been noted by Zhai et al. (2020)). Note that we could leverage these pre-trained weights because of CNGRADcan start from a pre-trained model by initializing the extra affine layers to the identity. Finally, we use $\sigma' = \sqrt{\sigma^2 - \nu^2} \approx 0.23, 0.49, 0.995$ so that the points used in our tailoring loss come from $N(x, \sigma^2)$.

In table 3, we show results on Imagenet where we improve the average certification radius by $5.1\%, 13.8\%, 19.6\%$ respectively. In table 6, in the appendix, we show results on CIFAR-10 where we improve the average certification radius by $8.6\%, 10.4\%, 19.2\%$. We chose to meta-tailor this randomized smoothing method because it represents a strong standard in certified adversarial defenses, but it is important to note that there have been advances on this method that sometimes achieve better results than those we present here (Zhai et al., 2020; Salman et al., 2019), see appendix H. However, it is likely that these methods could also be improved through meta-tailoring.

These experiments only scratch the surface of what tailoring allows for adversarial defenses: usually, the adversary looks at the model and gets to pick a particularly bad perturbation $x + \delta$. With tailoring, the model responds, by changing to weights $\theta_{x+\delta}$. This leads to a game, where both weights and inputs are perturbed, similar to: $\max_{|\delta| < \epsilon_x} \min_{|\Delta| < \epsilon_\theta} \mathcal{L}^{\text{sup}}(f_{\theta+\Delta}(x + \delta), y)$. However, since we don't get to observe $y$; we optimize the weight perturbation by minimizing $\mathcal{L}^{\text{tailor}}$ instead.

## 5.3 TAILORING TO IMPROVE THE PERFORMANCE OF MODEL-BASED RL

In model-based reinforcement learning (MBRL) we consider a Markov decision process (MDP) with state-space $\mathcal{S}$, action-space $\mathcal{A}$ and *unknown* transition distribution $p(s'|s, a)$. Here, we assume access to a *known* reward function $r(s, a)$, although this assumption could be relaxed. In MBRL we learn a deterministic transition model $T_\theta(s, a) \to s'$ from past experience and use it to either learn a policy or (as done here) to plan good actions by maximizing the reward over some planning horizon $H$:

$$a^*_{t...t+H} = \arg\max_{a_{t...t+H-1}} \sum_{h=1}^{H} r(\widehat{s_{t+h}}) = \arg\max_{a_{t...t+H-1}} \sum_{h=1}^{H} r(T_\theta(s_t, a_{t...t+h-1}))$$

where $T_\theta(s_t, a_{t...t+h-1}) = T_\theta(\dots T_\theta(T_\theta(s_t, a_t), a_{t+1}), \dots, a_{t+h-1})$, i.e. applying $T_\theta$ $h$ times. We also often apply receding-horizon control, where we plan with horizon $H$, but then only apply the first action $a_t$, observe the next state $a_{t+1}$ and re-plan again. However, in this setting we are optimizing the output of the transition model with respect to (part of) the input: the actions. Similar to adversarial examples in the previous section, this often results in the action optimisation finding regions of the input-space where the model makes overly-optimistic predictions. In order to improve performance it has been previously proposed to regularize the confidence of the models (Ha & Schmidhuber, 2018), to make them uncertainty-aware by learning an ensemble of models (Nagabandi

et al., 2020), or to make pessimistic predictions (Kidambi et al., 2020), among many other approaches. Similar to the adversarial examples experiments, meta-tailoring opens new simple ways of making transition models more robust.

In this section, we explain a simple way of using meta-tailoring to improve transition models in MBRL. In particular, we can learn an action-independent model $M(s_t) \to p(s_{t+1})$ that outputs a probability distribution for the next state given the current state, independently of the action $a_t$. This model presents a trade-off with respect to the original model $T(s, a)$: on the one hand, it is intrinsically uncertain since it lacks part of the information (the actions); we thus output a probability distribution in the form of a mixture of Gaussians to model this uncertainty. On the other hand, it has the advantage of not being "hackable" by the action-selection process, making the planner unable to find overly confident regions.

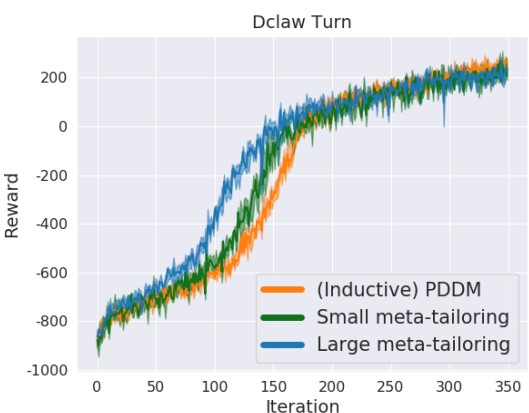

Figure 4: Meta-tailoring improves the performance of MBRL in dclaw, a complex manipulation problem, especially for low-data regimes where the transition model is imperfect and the prior is thus useful. We plot one confidence interval for the median reward bootstrapped from 16 different runs per configuration. Further increasing the tailoring learning rate results in (meta-)training instabilities because the inner loop is highly variable.

More concretely, we first train an action-independent prediction model to maximize the log-likelihood $M(s_t)(s_{t+1})$. Then, when making predictions with $T_\theta(s_t, a_t)$ we use the action-independent log-likelihood of the action-dependent predictions as the tailoring loss: $-M(s_t)(T_\theta(s_t, a_t))$, with a minus sign because we want to maximize the log-likelihood. From this we obtain tailored parameters $\theta_{(s_t, a_t)}$ and use them to make the final prediction. In meta-tailoring fashion we train these final predictions to be close to the truth $T_{\theta_{(s_t, a_t)}}(s_t, a_t) \approx s_{t+1}$. In practice, we use the first-order version of CNGRAD.

We use this idea and build on PDDM (Nagabandi et al., 2020), which recently showed great results in model-based reinforcement learning for robotic manipulation problems. Since they use an ensemble of deep networks $T_i(s_t, a_t)$ (to make robust predictions), we independently meta-tailor each network to leverage a single action-independent learned prior $M(s_t)$, leaving the rest of PDDM the same. In figure 4, we observe that meta-tailoring improves the performance of the overall predictive model, especially during early parts of training when the action-conditioned model has not become accurate. As expected, once the action-conditioned model is good enough (to get positive reward) meta-tailoring is not as useful as we do not need the action-independent prior anymore. For more experimental details, see appendix I.

## 6 CONCLUSION

We have presented *tailoring*, a simple way of embedding a powerful class of inductive biases into models, by minimizing unsupervised objectives at prediction time. We have leveraged the generality of auxiliary losses and improved them in two key ways: first, we eliminate the generalization gap on the auxiliary loss by optimizing it on the query point instead of on a training set; second, we change the optimization to minimize only the task loss in the outer optimization, and the tailoring loss in the inner optimization. This results in the whole network optimizing the only objective we care about, instead of a proxy loss. Finally, we have formalized these intuitions by proving the benefits of meta-tailoring under mild assumptions.

The framework is broadly applicable, as one can vary the model, the unsupervised loss and the task loss. We have shown its applicability in three diverse domains: physics prediction time-series, contrastive learning, and adversarial robustness. We also provide a simple algorithm, CNGRAD, to make meta-tailoring practical with little additional code. Currently, most unsupervised or self-supervised objectives are optimized in task-agnostic ways; instead, meta-tailoring provides a generic way to make them especially useful for particular applications. It does so by learning how to best leverage the unsupervised loss to perform well on the final task we care about.

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

## A  ON THE CONDITIONS IN THEOREM 2 AND COROLLARY 1

Assumption 1 is satisfied by using common activation functions such as sigmoid and hyperbolic tangent, as well as *softplus*, which is defined as $\sigma_\alpha(x) = \ln(1+\exp(\alpha x))/\alpha$ and satisfies Assumption 1 with any hyperparameter $\alpha \in \mathbb{R}_{>0}$. The softplus activation function can approximate the ReLU function to any desired accuracy: i.e., $\sigma_\alpha(x) \to \mathrm{relu}(x)$ as $\alpha \to \infty$, where $\mathrm{relu}$ represents the ReLU function.

In Theorem 2 and Corollary 1, the condition $\|g^{(i)}(x)\|_2^2 - g^{(i)}(x)^\top g^{(j)}(x) > 0$ (for all $i \neq j$) can be easily satisfied, for example, by choosing $g^{(1)}, \ldots, g^{(n_g)}$ to produce normalized and distinguishable argumented inputs for each prediction point $x$ at prediction time. To see this, with normalization $\|g^{(i)}(x)\|_2^2 = \|g^{(j)}(x)\|_2^2$, the condition is satisfied if $\|g^{(i)}(x) - g^{(j)}(x)\|_2^2 > 0$ for $i \neq j$ since $\frac{1}{2}\|g^{(i)}(x) - g^{(j)}(x)\|_2^2 = \|g^{(i)}(x)\|_2^2 - g^{(i)}(x)^\top g^{(j)}(x)$.

In general, the normalization is not necessary for the condition to hold; e.g., orthogonality on $g^{(i)}(x)$ and $g^{(j)}(x)$ along with $g^{(i)}(x) \neq 0$ satisfies it without the normalization.

## B  UNDERSTANDING THE EXPECTED META-TAILORING CONTRASTIVE LOSS

To analyze meta-tailoring for contrastive learning, we focus on the binary classification loss of the form $\mathcal{L}^{\sup}(f_\theta(x), y) = \ell_{\mathrm{cont}}(f_\theta(x)_y - f_\theta(x)_{y'=\neg y})$ where $\ell_{\mathrm{cont}}$ is convex and $\ell_{\mathrm{cont}}(0) = 1$. With this, the objective function $\theta \mapsto \mathcal{L}^{\sup}(f_\theta(x), y)$ is still non-convex in general. For example, the standard hinge loss $\ell_{\mathrm{cont}}(z) = \max\{0, 1 - z\}$ and the logistic loss $\ell_{\mathrm{cont}}(z) = s\log_2(1 + \exp(z))$ satisfy this condition.

We first define the meta-tailoring contrastive loss $\mathcal{L}_{\mathrm{cont}}(x, \theta)$ in detail. In meta-tailoring contrastive learning, we choose the probability measure of positive example $x^+ \sim \mu_{x^+}(x)$ and the probability measure of negative example $x^-, y^- \sim \mu_{x^-, y^-}(x)$, both of which are tailored for each input point $x$ at prediction time. These choices induce the marginal distributions for the negative examples $y^- \sim \mu_{y^-}(x)$ and $x^- \sim \mu_{x^-}(x)$, as well as the unknown probability of $y^- = y$ defined by $\rho_y(\mu_{y^-}(x)) = \mathbb{E}_{y^- \sim \mu_{y^-}(x)}(\mathbb{1}\{y^- = y\})$. Define the lower and upper bound on the probability of $y^- = y$ as $\underline{\rho}(x) \leq \rho_y(\mu_{y^-}(x)) \leq \bar{\rho}(x) \in [0, 1)$.

Then, the first pre-meta-tailoring contrastive loss can be defined by

$$\mathcal{L}_{\mathrm{cont}}^{x^+, x^-}(x, \theta) = \mathbb{E}_{\substack{x^+ \sim \mu_{x^+}(x), \\ x^- \sim \mu_{x^-}(x)}}[\ell_{\mathrm{cont}}(h_\theta(x)^\top(h_\theta(x^+) - h_\theta(x^-)))],$$

where $h_\theta(x) \in \mathbb{R}^{m_H+1}$ represents the output of the last hidden layer, including a constant neuron corresponding the bias term of the last output layer (if there is no bias term, $h_\theta(x) \in \mathbb{R}^{m_H}$). For every $z \in \mathbb{R}^{2\times(m_H+1)}$, define $\psi_{x,y,y^-}(z) = \ell_{\mathrm{cont}}((z_y - z_{y^-})h_\theta(x))$, where $z_y \in \mathbb{R}^{1\times m_H}$ is the $y$-th row vector of $z$. We define the second pre-meta-tailoring contrastive loss by

$$\mathcal{L}_{\mathrm{cont}}^{x^+, x^-, y^-}(x, \theta) = \max_y \mathbb{E}_{y^- \sim \mu_{y^-}(x)}[\psi_{x,y,y^-}(\theta^{(H+1)}) - \psi_{x,1,2}([u_h^+, u_h^-]^\top)],$$

where $u_h^+ = \mathbb{E}_{x^+ \sim \mu_{x^+}(x)}[h_\theta(x^+)]$ and $u_h^- = \mathbb{E}_{x^- \sim \mu_{x^-}(x)}[h_\theta(x^-)]$. Here, we decompose $\theta$ as $\theta = (\theta^{(1:H)}, \theta^{(H+1)})$, where $\theta^{(H+1)} = [W^{(H+1)}, b^{(H+1)}] \in \mathbb{R}^{m_y \times (m_H+1)}$ represents the parameters at the last output layer, and $\theta^{(1:H)}$ represents all others.

Then, the meta-tailoring contrastive loss is defined by

$$\mathcal{L}_{\mathrm{cont}}(x, \theta) = \frac{1}{1 - \bar{\rho}(x)}\left(\mathcal{L}_{\mathrm{cont}}^{x^+, x^-}(x, \theta) + \mathcal{L}_{\mathrm{cont}}^{x^+, x^-, y^-}(x, \theta) - \underline{\rho}(x)\right).$$

Theorem 3 states that for any $\theta^{(1:H)}$, the convex optimization of $\mathcal{L}_{\mathrm{cont}}^{x^+, x^-}(x, \theta) + \mathcal{L}_{\mathrm{cont}}^{x^+, x^-, y^-}(x, \theta)$ over $\theta^{(H+1)}$ can achieve the value of $\mathcal{L}_{\mathrm{cont}}^{x^+, x^-}(x, \theta)$ without the value of $\mathcal{L}_{\mathrm{cont}}^{x^+, x^-, y^-}(x, \theta)$, allowing us to focus on the first term $\mathcal{L}_{\mathrm{cont}}^{x^+, x^-}(x, \theta)$, for some choice of $\mu_{x^-, y^-}(x)$ and $\mu_{x^+}(x)$.

**Theorem 3.** *For any $\theta^{(1:H)}, \mu_{x^-,y^-}(x)$ and $\mu_{x^+}(x)$, the function $\theta^{(H+1)} \mapsto \mathcal{L}_{\mathrm{cont}}^{x^+,x^-}(x,\theta) + \mathcal{L}_{\mathrm{cont}}^{x^+,x^-,y^-}(x,\theta)$ is convex. Moreover, there exists $\mu_{x^-,y^-}(x)$ and $\mu_{x^+}(x)$ such that, for any $\theta^{(1:H)}$ and any $\bar{\theta}^{(H+1)}$,*

$$\inf_{\theta^{(H+1)}\in\mathbb{R}^{m_y\times(m_H+1)}} \mathcal{L}_{\mathrm{cont}}^{x^+,x^-}(x,\theta) + \mathcal{L}_{\mathrm{cont}}^{x^+,x^-,y^-}(x,\theta) \le \mathcal{L}_{\mathrm{cont}}^{x^+,x^-}(x,\theta^{(1:H)},\bar{\theta}^{(H+1)}).$$

## C    PROOFS

In order to have concise proofs, we introduce additional notations while keeping track of dependent variables more explicitly. Since $h_\theta$ only depends on $\theta^{(1:H)}$, let us write $h_{\theta^{(1:H)}} = h_\theta$. Similarly, $\mathcal{L}_{\mathrm{cont}}(x,\theta^{(1:H)}) = \mathcal{L}_{\mathrm{cont}}(x,\theta)$. Let $\theta(x) = \theta_x$ and $\theta(x,S) = \theta_{x,S}$. Define $\mathcal{L} = \mathcal{L}^{\mathrm{sup}}$.

### C.1    PROOF OF THEOREM 2

*Proof of Theorem 2.* The output of fully-connected neural networks for an input $x$ with a parameter vector $\theta = (w,\gamma,\beta)$ can be represented by $f_\theta(x) = W^{(H+1)}h^{(H)}(x) + b^{(H+1)}$ where $W^{(H+1)} \in \mathbb{R}^{1\times m_H}$ and $b^{(H+1)} \in \mathbb{R}$ are the weight matrix and the bias term respectively at the last layer, and $h^{(H)}(x) \in \mathbb{R}^{m_H}$ represents the output of the last hidden layer. Here, $m_l$ represents the number of neurons at the $l$-th layer, and $h^{(l)}(x) = \gamma^{(l)}(\sigma(W^{(l)}h^{(l-1)}(x)+b^{(l)})) - \beta^{(l)} \in \mathbb{R}^{m_l}$ for $l = 1,\ldots,H$, with trainable parameters $\gamma^{(l)}, \beta^{(l)} \in \mathbb{R}^{m_l}$, where $h^{(0)}(x) = x$. Let $z^{(l)}(x) = \sigma(W^{(l)}h^{(l-1)}(x) + b^{(l)})$.

Then, by rearranging the definition of the output of the neural networks,

$$\begin{aligned}
f_\theta(x) &= W^{(H+1)}h^{(H)}(x) + b^{(H+1)} \\
&= \left(\sum_{k=1}^{m_H} W_k^{(H+1)}\gamma_k^{(H)}z^{(H)}(x)_k + W_k^{(H+1)}\beta_k^{(H)}\right) + b^{(H+1)} \\
&= [W^{(H+1)}\circ z^{(H)}(x)^\top, W^{(H+1)}]\begin{bmatrix}\gamma^{(H)}\\\beta^{(H)}\end{bmatrix} + b^{(H+1)}.
\end{aligned}$$

Thus, we can write

$$\begin{bmatrix}f_\theta(g^{(1)}(x))\\\vdots\\f_\theta(g^{(n_g)}(x))\end{bmatrix} = M_w\begin{bmatrix}\gamma^{(H)}\\\beta^{(H)}\end{bmatrix} + b^{(H+1)}\mathbf{1}_{n_g} \in \mathbb{R}^{n_g}, \tag{2}$$

where

$$M_w = \begin{bmatrix}W^{(H+1)}\circ z^{(H)}(g^{(1)}(x))^\top, W^{(H+1)}\\\vdots\\W^{(H+1)}\circ z^{(H)}(g^{(n_g)}(x))^\top, W^{(H+1)}\end{bmatrix} \in \mathbb{R}^{n_g\times 2m_H},$$

and $\mathbf{1}_{n_g} = [1,1,\ldots,1]^\top \in \mathbb{R}^{n_g}$.

Using the above equality, we show an exitance of a $(\gamma,\beta)$ such that $\mathcal{L}^{\mathrm{tailor}}(x,\bar{w},\gamma,\beta) = \inf_{w,\gamma,\beta}\mathcal{L}^{\mathrm{tailor}}(x,\theta)$ for any $x \in \mathcal{X} \subseteq \mathbb{R}^{m_x}$ and any $\bar{w} \notin \mathcal{W}$ where Lebesgue measure of $\mathcal{W} \subset \mathbb{R}^d$ is zero. To do so, we first fix $\gamma_k^{(l)} = 1$ and $\beta_k^{(l)} = 0$ for $l = 1,\ldots,H-1$, with which $h^{(l)}(x) = z^{(l)}(x)$ for $l = 1,\ldots,H-1$.

Define $\varphi(w) = \det(M_w M_w^\top)$, which is analytic since $\sigma$ is analytic. Furthermore, we have that $\{w \in \mathbb{R}^d : M_w \text{ has rank less than } n_g\} = \{w \in \mathbb{R}^d : \varphi(w) = 0\}$, since the rank of $M_w$ and the rank of the Gram matrix are equal. Since $\varphi$ is analytic, if $\varphi$ is not identically zero ($\varphi \ne 0$), the Lebesgue measure of its zero set $\{w \in \mathbb{R}^d : \varphi(w) = 0\}$ is zero (Mityagin, 2015). Therefore, if $\varphi(w) \ne 0$ for some $w \in \mathbb{R}^d$, the Lebesgue measure of the set $\{w \in \mathbb{R}^d : M_w \text{ has rank less than } n_g\}$ is zero.

Accordingly, we now constructs a $w \in \mathbb{R}^d$ such that $\varphi(w) \ne 0$. Set $W^{(H+1)} = \mathbf{1}_{m_H}^\top$. Then,

$$M_w = [\bar{M}_w, \mathbf{1}_{n_g,m_H}] \in \mathbb{R}^{n_g\times m_H}.$$

where

$$\bar{M}_w = \begin{bmatrix} z^{(H)}(g^{(1)}(x))^\top \\ \vdots \\ z^{(H)}(g^{(n_g)}(x)(x))^\top \end{bmatrix} \in \mathbb{R}^{n_g \times m_H}$$

and $\mathbf{1}_{n_g,m_H} \in \mathbb{R}^{n_g \times m_H}$ with $(\mathbf{1}_{n_g,m_H})_{ij} = 1$ for all $i, j$. For $l = 1, \ldots, H$, define

$$G^{(l)} = \begin{bmatrix} z^{(l)}(g^{(1)}(x))^\top \\ \vdots \\ z^{(l)}(g^{(n_g)}(x))^\top \end{bmatrix} \in \mathbb{R}^{n_g \times m_l}.$$

Then, for $l = 1, \ldots, H$,

$$G^{(l)} = \sigma(G^{(l-1)}(W^{(l)})^\top + \mathbf{1}_{n_g}(b^{(l)})^\top),$$

where $\sigma$ is applied element-wise (by overloading of the notation $\sigma$), and

$$(\bar{M}_w)_{ik} = (G^{(H)})_{ik}.$$

From the assumption $g(x)$, there exists $c > 0$ such that $\|g^{(i)}(x)\|_2^2 - \langle g^{(i)}(x), g^{(j)}(x) \rangle > c$ for all $i \neq j$. From Assumption 1, there exists $c'$ such that $\sigma_+ - \sigma_- > c'$. Using these constants, set $W_i^{(1)} = \alpha^{(1)}g^{(i)}(x)^\top$ and $b_i^{(1)} = c\alpha^{(1)}/2 - \alpha^{(1)}\|g^{(i)}(x)\|_2^2$ for $i = 1, \ldots, n_g$, where $W_i^{(1)}$ represents the $i$-th row of $W^{(1)}$. Moreover, set $W_{1:n_g,1:n_g}^{(l)} = \alpha^{(l)}I_{n_g}$ and $b_k^{(l)} = c'\alpha^{(l)}/2 - \alpha^{(l)}\sigma_+$ for all $k$ and $l = 2, \ldots, H$, where $W_{1:n_g,1:n_g}^{(l)}$ is the fist $n_g \times n_g$ block matrix of $W^{(1)}$ and $I_{n_g}$ is the $n_g \times n_g$ identity matrix. Set all other weights and bias to be zero. Then, for any $i \in \{1, \ldots, n_g\}$,

$$(G^{(1)})_{ii} = \sigma(c\alpha^{(1)}/2),$$

and for any $k \in \{1, \ldots, n_g\}$ with $k \neq i$,

$$(G^{(1)})_{ik} = \sigma(\alpha^{(1)}(\langle g^{(i)}(x), g^{(k)}(x) \rangle - \|g^{(k)}(x)\|_2^2 + c/2)) \leq \sigma(-c\alpha^{(1)}/2).$$

Since $\sigma(c\alpha^{(1)}/2) \to \sigma_+$ and $\sigma(-c\alpha^{(1)}/2) \to \sigma_-$ as $\alpha^{(1)} \to \infty$, with $\alpha^{(1)}$ sufficiently large, we have that $\sigma(c\alpha^{(1)}/2) - \sigma_+ + c'/2 \geq c_1^{(2)}$ and $\sigma(-c\alpha^{(1)}/2) - \sigma_+ + c'/2 \leq -c_2^{(2)}$ for some $c_1^{(2)}, c_2^{(2)} > 0$. Note that $c_1^{(2)}$ and $c_2^{(2)}$ depends only on $\alpha^{(1)}$ and does not depend on any of $\alpha^{(2)}, \ldots, \alpha^{(H)}$. Therefore, with $\alpha^{(1)}$ sufficiently large,

$$(G^{(2)})_{ii} = \sigma(\alpha^{(2)}(\sigma(c\alpha^{(1)}/2) - \sigma_+ + c'/2)) \geq \sigma(\alpha^{(2)}c_1^{(2)}),$$

and

$$(G^{(2)})_{ik} \leq \sigma(\alpha^{(2)}(\sigma(-c\alpha^{(1)}/2) - \sigma_+ + c'/2)) \leq \sigma(-\alpha^{(2)}c_2^{(2)}).$$

Repeating this process with Assumption 1, we have that with $\alpha^{(1)}, \ldots, \alpha^{(H-1)}$ sufficiently large,

$$(G^{(H)})_{ii} \geq \sigma(\alpha^{(H)}c_1^{(H)}),$$

and

$$(G^{(H)})_{ik} \leq \sigma(-\alpha^{(H)}c_2^{(H)}).$$

Here, $(G^{(H)})_{ii} \to \sigma_+$ and $(G^{(H)})_{ik} \to \sigma_-$ as $\alpha^{(H)} \to \infty$. Therefore, with $\alpha^{(1)}, \ldots, \alpha^{(H)}$ sufficiently large, for any $i \in \{1, \ldots, n_g\}$,

$$\left|(\bar{M}_w)_{ii} - \sigma_-\right| > \sum_{k \neq i} \left|(\bar{M}_w)_{ik} - \sigma_-\right|. \tag{3}$$

The inequality equation 3 means that the matrix $\bar{M}_w' = [(\bar{M}_w)_{ij} - \sigma_-]_{1 \leq i,j \leq n_g} \in \mathbb{R}^{n_g \times n_g}$ is strictly diagonally dominant and hence is nonsingular with rank $n_g$. This implies that the matrix $[\bar{M}_w', \mathbf{1}_{n_g}] \in \mathbb{R}^{n_g \times (n_g+1)}$ has rank $n_g$. This then then implies that the matrix $\tilde{M}_w = [[(\bar{M}_w)_{ij}]_{1 \leq i,j \leq n_g}, \mathbf{1}_{n_g}] \in \mathbb{R}^{n_g \times (n_g+1)}$ has rank $n_g$, since the elementary matrix operations preserve the matrix rank. Since the set of all columns of $M_w$ contains all columns of $\tilde{M}_w$, this implies that $M_w$ has rank $n_g$ and $\varphi(w) \neq 0$ for this constructed particular $w$.

Therefore, the Lebesgue measure of the set $\mathcal{W} = \{w \in \mathbb{R}^d : \varphi(w) = 0\}$ is zero. If $w \notin \mathcal{W}$, $\{(f_{\bar{w},\bar{\gamma},\bar{\beta}}(g^{(1)}(x)), \ldots, f_{\bar{w},\bar{\gamma},\bar{\beta}}(g^{(n_g)}(x)) \in \mathbb{R}^{n_g} : \bar{\gamma}^{(l)}, \bar{\beta}^{(l)} \in \mathbb{R}^{ml}\} = \mathbb{R}^{n_g}$, since $M_w$ has rank $n_g$ in equation 2 for some $\bar{\gamma}^{(l)}, \bar{\beta}^{(l)}$ for $l = 1, \ldots, H-1$ as shown above. Thus, for any $\bar{w} \notin \mathcal{W}$ and for any $(w, \gamma, \beta)$, there exists $(\bar{\gamma}, \bar{\beta})$ such that

$$(f_{w,\gamma,\beta}(g^{(1)}(x)), \ldots, f_{w,\gamma,\beta}(g^{(n_g)}(x)) = (f_{\bar{w},\bar{\gamma},\bar{\beta}}(g^{(1)}(x)), \ldots, f_{\bar{w},\bar{\gamma},\bar{\beta}}(g^{(n_g)}(x))$$

which implies the desired statement.

$\square$

## C.2 PROOF OF COROLLARY 1

*Proof of Corollary 1.* Since non-degenerate Gaussian measure with any mean and variance is absolutely continuous with respect to Lebesgue measure, Theorem 2 implies the statement of this corollary.

$\square$

## C.3 PROOF OF THEOREM 1

The following lemma provides an upper bound on the expected loss via expected meta-tailoring contrastive loss.

**Lemma 4.** *For every $\theta$,*

$$\mathbb{E}_{x,y}[\mathcal{L}(f_\theta(x), y)] \leq \mathbb{E}_x\left[\frac{1}{1-\bar{\rho}(x)}\left(\mathcal{L}_{\text{cont}}^{x^+,x^-}(x, \theta^{(1:H)}) + \mathcal{L}_{\text{cont}}^{x^+,x^-,y^-}(x, \theta) - \underline{\rho}(x)\right)\right]$$

*Proof of Lemma 4.* Using the notation $\rho = \rho_y(\mu_{y^-}(x))$,

$$\mathbb{E}_{x,y}[\mathcal{L}(f_\theta(x), y)]$$

$$= \mathbb{E}_{x,y}\left[\frac{1}{1-\rho}((1-\rho)\mathcal{L}(f_\theta(x), y) \pm \rho)\right]$$

$$= \mathbb{E}_{x,y}\left[\frac{1}{1-\rho}((1-\rho)\ell_{\text{cont}}(f_\theta(x)_y - f_\theta(x)_{y^- \neq y}) + \rho\ell_{\text{cont}}(f_\theta(x)_y - f_\theta(x)_{y^- = y}) - \rho)\right]$$

$$= \mathbb{E}_{x,y}\left[\frac{1}{1-\rho}\left(\mathbb{E}_{y^- \sim \mu_{y^-}(x)}[\ell_{\text{cont}}(f_\theta(x)_y - f_\theta(x)_{y^-})] - \rho\right)\right]$$

$$= \mathbb{E}_{x,y}\left[\frac{1}{1-\rho}\left(\mathbb{E}_{y^- \sim \mu_{y^-}(x)}[\psi_{x,y,y^-}(\theta^{(H+1)})] - \rho\right)\right]$$

$$\leq \mathbb{E}_{x,y}\left[\frac{1}{1-\rho}\left(\psi_{x,1,2}([u_h^+, u_h^-]^\top) + \mathcal{L}_{\text{cont}}^{x^+,x^-,y^-}(x, \theta) - \rho\right)\right]$$

$$\leq \mathbb{E}_{x,y}\left[\frac{1}{1-\rho}\left(\mathcal{L}_{\text{cont}}^{x^+,x^-}(x, \theta^{(1:H)}) + \mathcal{L}_{\text{cont}}^{x^+,x^-,y^-}(x, \theta) - \rho\right)\right]$$

where the third line follows from the definition of $\mathcal{L}(f_\theta(x), y)$ and $\ell_{\text{cont}}(f_\theta(x)_y - f_\theta(x)_{y'=y}) = \ell_{\text{cont}}(0) = 1$, the forth line follows from the definition of $\rho$ and the expectation $\mathbb{E}_{y^- \sim \mu_{y^-}(x)}$, the fifth line follows from $f_\theta(x)_y = \theta_y^{(H+1)} h_{\theta(1:H)}(x)$ and $f_\theta(x)_{y^-} = \theta_{y^-}^{(H+1)} h_{\theta(1:H)}(x)$, the sixth line follows from the definition of $\mathcal{L}_{\text{cont}}^{x^+,x^-,y^-}$. The last line follows from the convexity of $\ell_{\text{cont}}$ and Jensen's inequality: i.e.,

$$\psi_{x,1,2}([u_h^+, u_h^-]^\top)$$

$$= \ell_{\text{cont}}(\mathbb{E}_{x^+ \sim \mu_{x^+}(x)}\mathbb{E}_{x^- \sim \mu_{x^-}(x)}[(h_{\theta(1:H)}(x^+) - h_{\theta(1:H)}(x^-))^\top h_{\theta(1:H)}(x)])$$

$$\leq \mathbb{E}_{x^+ \sim \mu_{x^+}(x)}\mathbb{E}_{x^- \sim \mu_{x^-}(x)}\ell_{\text{cont}}((h_{\theta(1:H)}(x^+) - h_{\theta(1:H)}(x^-))^\top h_{\theta(1:H)}(x)).$$

Therefore,

$$\mathbb{E}_{x,y}[\mathcal{L}(f_\theta(x), y)]$$

$$\leq \mathbb{E}_{x,y}\left[\frac{1}{1 - \rho_y(\mu_{y^-}(x))}\left(\mathcal{L}_{\text{cont}}^{x^+,x^-}(x, \theta^{(1:H)}) + \mathcal{L}_{\text{cont}}^{x^+,x^-,y^-}(x, \theta) - \rho_y(\mu_{y^-}(x))\right)\right]$$

$$\leq \mathbb{E}_x\left[\frac{1}{1 - \bar{\rho}(x)}\left(\mathcal{L}_{\text{cont}}^{x^+,x^-}(x, \theta^{(1:H)}) + \mathcal{L}_{\text{cont}}^{x^+,x^-,y^-}(x, \theta) - \underline{\rho}(x)\right)\right]$$

where we used $\underline{\rho}(x) \leq \rho_y(\mu_{y^-}(x)) \leq \bar{\rho}(x) \in [0, 1)$. $\qquad\square$

**Lemma 5.** *Let $S \mapsto f_{\theta(x,S)}(x)$ be an uniformly $\zeta$-stable tailoring algorithm. Then, for any $\delta > 0$, with probability at least $1 - \delta$ over an i.i.d. draw of $n$ i.i.d. samples $S = ((x_i, y_i))_{i=1}^n$, the following holds:*

$$\mathbb{E}_{x,y}[\mathcal{L}(f_{\theta(x,S)}(x), y)] \leq \frac{1}{n}\sum_{i=1}^n \mathcal{L}(f_{\theta(x_i,S)}(x_i), y_i) + \frac{\zeta}{n} + (2\zeta + c)\sqrt{\frac{\ln(1/\delta)}{2n}}.$$

*Proof of Lemma 5.* Define $\varphi_1(S) = \mathbb{E}_{x,y}[\mathcal{L}(f_{\theta(x,S)}(x), y)]$ and $\varphi_2(S) = \frac{1}{n}\sum_{i=1}^n \mathcal{L}(f_{\theta(x_i,S)}(x_i), y_i)$, and $\varphi(S) = \varphi_1(S) - \varphi_2(S)$. To apply McDiarmid's inequality to $\varphi(S)$, we compute an upper bound on $|\varphi(S) - \varphi(S')|$ where $S$ and $S'$ be two training datasets differing by exactly one point of an arbitrary index $i_0$; i.e., $S_i = S_i'$ for all $i \neq i_0$ and $S_{i_0} \neq S_{i_0}'$, where $S' = ((x_i', y_i'))_{i=1}^n$. Let $\tilde{\zeta} = \frac{\zeta}{n}$ Then,

$$|\varphi(S) - \varphi(S')| \leq |\varphi_1(S) - \varphi_1(S')| + |\varphi_2(S) - \varphi_2(S')|.$$

For the first term, using the $\zeta$-stability,

$$|\varphi_1(S) - \varphi_1(S')| \leq \mathbb{E}_{x,y}[|\mathcal{L}(f_{\theta(x,S)}(x), y) - \mathcal{L}(f_{\theta(x,S')}(x), y)|]$$
$$\leq \tilde{\zeta}.$$

For the second term, using $\zeta$-stability and the upper bound $c$ on per-sample loss,

$$|\varphi_2(S) - \varphi_2(S')| \leq \frac{1}{n}\sum_{i \neq i_0}|\mathcal{L}(f_{\theta(x_i,S)}(x_i), y_i) - \mathcal{L}(f_{\theta(x_i,S')}(x_i), y_i)| + \frac{c}{n}$$

$$\leq \frac{(n-1)\tilde{\zeta}}{n} + \frac{c}{n} \leq \tilde{\zeta} + \frac{c}{n}.$$

Therefore, $|\varphi(S) - \varphi(S')| \leq 2\tilde{\zeta} + \frac{c}{n}$. By McDiarmid's inequality, for any $\delta > 0$, with probability at least $1 - \delta$,

$$\varphi(S) \leq \mathbb{E}_S[\varphi(S)] + (2\zeta + c)\sqrt{\frac{\ln(1/\delta)}{2n}}.$$

The reset of the proof bounds the first term $\mathbb{E}_S[\varphi(S)]$. By the linearity of expectation,

$$\mathbb{E}_S[\varphi(S)] = \mathbb{E}_S[\varphi_1(S)] - \mathbb{E}_S[\varphi_1(S)].$$

For the first term,

$$\mathbb{E}_S[\varphi_1(S)] = \mathbb{E}_{S,x,y}[\mathcal{L}(f_{\theta(x,S)}(x), y)].$$

For the second term, using the linearity of expectation,

$$\mathbb{E}_S[\varphi_2(S)] = \mathbb{E}_S\left[\frac{1}{n}\sum_{i=1}^n \mathcal{L}(f_{\theta(x_i,S)}(x_i), y_i)\right]$$

$$= \frac{1}{n}\sum_{i=1}^n \mathbb{E}_S[\mathcal{L}(f_{\theta(x_i,S)}(x_i), y_i)]$$

$$= \frac{1}{n}\sum_{i=1}^n \mathbb{E}_{S,x,y}[\mathcal{L}(f_{\theta(x,S_{x,y}^i)}(x), y)],$$

where $S^i$ is a sample of $n$ points such that $(S^i_{x,y})_j = S_j$ for $j \neq i$ and $(S^i_{x,y})_i = (x,y)$. By combining these, using the linearity of expectation and $\zeta$-stability,

$$\mathbb{E}_S[\varphi(S)] = \frac{1}{n}\sum_{i=1}^n \mathbb{E}_{S,x,y}[\mathcal{L}(f_{\theta(x,S)}(x),y) - \mathcal{L}(f_{\theta(x,S^i_{x,y})}(x),y)]$$

$$\leq \frac{1}{n}\sum_{i=1}^n \mathbb{E}_{S,x,y}[|\mathcal{L}(f_{\theta(x,S)}(x),y) - \mathcal{L}(f_{\theta(x,S^i_{x,y})}(x),y)|]$$

$$\leq \frac{1}{n}\sum_{i=1}^n \tilde{\zeta} = \tilde{\zeta}.$$

Therefore, $\mathbb{E}_S[\varphi(S)] \leq \tilde{\zeta}$.

$\square$

*Proof of Theorem 1.* For any $\theta$ and $\kappa \in [0,1]$,

$$\mathbb{E}_{x,y}[\mathcal{L}(f_\theta(x),y)] = \kappa\mathbb{E}_{x,y}[\mathcal{L}(f_\theta(x),y)] + (1-\kappa)\mathbb{E}_{x,y}[\mathcal{L}(f_\theta(x),y)].$$

Applying Lemma 4 for the first term and Lemma 5 yields the desired statement.

$\square$

### C.4 STATEMENT AND PROOF OF THEOREM 6

**Theorem 6.** *Let $\mathcal{F}$ be an arbitrary set of maps $x \mapsto f_{\theta_x}(x)$. Then, for any $\delta > 0$, with probability at least $1-\delta$ over an i.i.d. draw of $n$ i.i.d. samples $((x_i,y_i))_{i=1}^n$, the following holds: for all maps $(x \mapsto f_{\theta_x}(x)) \in \mathcal{F}$ and any $\kappa \in [0,1]$, we have that $\mathbb{E}_{x,y}[\mathcal{L}^{\mathrm{sup}}(f_{\theta_x}(x),y)] \leq \kappa\mathbb{E}_x[\mathcal{L}_{\mathrm{cont}}(x,\theta_x)] + (1-\kappa)\mathcal{J}'$, where $\mathcal{J}' = \frac{1}{n}\sum_{i=1}^n \mathcal{L}^{\mathrm{sup}}(f_{\theta_{x_i}}(x_i),y_i) + 2\mathcal{R}_n(\mathcal{L}^{\mathrm{sup}} \circ \mathcal{F}) + c\sqrt{(\ln(1/\delta))/(2n)}$.*

The following lemma is used along with Lemma 4 to prove the statement of this theorem.

**Lemma 7.** *Let $\mathcal{F}$ be an arbitrary set of maps $x \mapsto f_{\theta_x}(x)$. For any $\delta > 0$, with probability at least $1-\delta$ over an i.i.d. draw of $n$ i.i.d. samples $((x_i,y_i))_{i=1}^n$, the following holds: for all maps $(x \mapsto f_{\theta_x}(x)) \in \mathcal{F}$,*

$$\mathbb{E}_{x,y}[\mathcal{L}(f_{\theta_x}(x),y)] \leq \frac{1}{n}\sum_{i=1}^n \mathcal{L}(f_{\theta_{x_i}}(x_i),y_i) + 2\mathcal{R}_n(\mathcal{L} \circ \mathcal{F}) + c\sqrt{\frac{\ln(1/\delta)}{2n}}.$$

*Proof of Lemma 7.* Let $S = ((x_i,y_i))_{i=1}^n$ and $S' = ((x_i',y_i'))_{i=1}^n$. Define

$$\varphi(S) = \sup_{(x \mapsto f_{\theta_x}(x)) \in \mathcal{F}} \mathbb{E}_{x,y}[\mathcal{L}(f_{\theta_x}(x),y)] - \frac{1}{n}\sum_{i=1}^n \mathcal{L}(f_{\theta_{x_i}}(x_i),y_i).$$

To apply McDiarmid's inequality to $\varphi(S)$, we compute an upper bound on $|\varphi(S) - \varphi(S')|$ where $S$ and $S'$ be two training datasets differing by exactly one point of an arbitrary index $i_0$; i.e., $S_i = S_i'$ for all $i \neq i_0$ and $S_{i_0} \neq S_{i_0}'$. Then,

$$\varphi(S') - \varphi(S) \leq \sup_{(x \mapsto f_{\theta_x}(x)) \in \mathcal{F}} \frac{\mathcal{L}(f_{\theta(x_{i_0})}(x_{i_0}),y_{i_0}) - \mathcal{L}(f_{\theta(x_{i_0}')}(x_{i_0}'),y_{i_0}')}{n} \leq \frac{c}{n}.$$

Similarly, $\varphi(S) - \varphi(S') \leq \frac{c}{n}$. Thus, by McDiarmid's inequality, for any $\delta > 0$, with probability at least $1-\delta$,

$$\varphi(S) \leq \mathbb{E}_S[\varphi(S)] + c\sqrt{\frac{\ln(1/\delta)}{2n}}.$$

Moreover, with $f(x) = f_{\theta_x}(x)$,

$$\mathbb{E}_S[\varphi(S)] = \mathbb{E}_S\left[\sup_{f\in\mathcal{F}}\mathbb{E}_{S'}\left[\frac{1}{n}\sum_{i=1}^n \mathcal{L}(f_{\theta(x'_i)}(x'_i), y'_i)\right] - \frac{1}{n}\sum_{i=1}^n \mathcal{L}(f_{\theta_{x_i}}(x_i), y_i)\right]$$

$$\leq \mathbb{E}_{S,S'}\left[\sup_{f\in\mathcal{F}}\frac{1}{n}\sum_{i=1}^n (\mathcal{L}(f_{\theta(x'_i)}(x'_i), y'_i) - \mathcal{L}(f_{\theta_{x_i}}(x_i), y_i)\right]$$

$$\leq \mathbb{E}_{\xi,S,S'}\left[\sup_{f\in\mathcal{F}}\frac{1}{n}\sum_{i=1}^n \xi_i(\mathcal{L}(f_{\theta(x'_i)}(x'_i), y'_i) - \mathcal{L}(f_{\theta_{x_i}}(x_i), y_i))\right]$$

$$\leq 2\mathbb{E}_{\xi,S}\left[\sup_{f\in\mathcal{F}}\frac{1}{n}\sum_{i=1}^n \xi_i\mathcal{L}(f_{\theta_{x_i}}(x_i), y_i))\right]$$

where the fist line follows the definitions of each term, the second line uses the Jensen's inequality and the convexity of the supremum, and the third line follows that for each $\xi_i \in \{-1, +1\}$, the distribution of each term $\xi_i(\mathcal{L}(f_{\theta(x'_i)}(x'_i), y'_i) - \mathcal{L}(f_{\theta_{x_i}}(x_i), y_i))$ is the distribution of $(\mathcal{L}(f_{\theta(x'_i)}(x'_i), y'_i) - \mathcal{L}(f_{\theta_{x_i}}(x_i), y_i))$ since $\bar{S}$ and $\bar{S}'$ are drawn iid with the same distribution. The forth line uses the subadditivity of supremum. □

*Proof of Theorem 6.* For any $\theta$ and $\kappa \in [0, 1]$,
$$\mathbb{E}_{x,y}[\mathcal{L}(f_\theta(x), y)] = \kappa\mathbb{E}_{x,y}[\mathcal{L}(f_\theta(x), y)] + (1 - \kappa)\mathbb{E}_{x,y}[\mathcal{L}(f_\theta(x), y)].$$
Applying Lemma 4 for the first term and Lemma 7 yields the desired statement.

□

## C.5 PROOF OF THEOREM 3

*Proof of Theorem 3.* Let $\theta^{(1:H)}$ be fixed. We first prove the first statement for the convexity. The function $\theta^{(H+1)} \mapsto \psi_{x,y,y^-}(\theta^{(H+1)})$ is convex, since it is a composition of a convex function $\ell_{\text{cont}}$ and a affine function $z \mapsto (z_y - z_{y^-})h_{\theta^{(1:H)}}(x)$. The function $\theta^{(H+1)} \mapsto \mathbb{E}_{y^-\sim\mu_{y^-}(x)}[\psi_{x,y,y^-}(\theta^{(H+1)}) - \psi_{x,1,2}([u_h^+, u_h^-]^\top)]$ is convex since the expectation and affine translation preserves the convexity. Finally, $\theta^{(H+1)} \mapsto \mathcal{L}_{\text{cont}}^{x^+,x^-,y^-}(x, \theta^{(1:H)}, \theta^{(H+1)})$ is convex since it is the piecewise maximum of the convex functions
$$\theta^{(H+1)} \mapsto \mathbb{E}_{y^-\sim\mu_{y^-}(x)}[\psi_{x,y,y^-}(\theta^{(H+1)}) - \psi_{x,1,2}([u_h^+, u_h^-]^\top)]$$
for each $y$.

We now prove the second statement of the theorem for the inequality. Let us write $\mu_{x^+} = \mu_{x|y}$ and $\mu_{x^-} = \mu_{x|y^-}$. Let $U = [u_1, u_2]^\top \in \mathbb{R}^{m_y\times(m_H+1)}$ where $u_y = \mathbb{E}_{x\sim\mu_{x|y}}[h_{\theta^{(1:H)}}(x)]$ for $y \in \{1, 2\}$. Then,
$$u_h^+ = \mathbb{E}_{x^+\sim\mu_{x^+}(x)}[h_{\theta^{(1:H)}}(x^+)] = \mathbb{E}_{x^+\sim\mu_{x|y}}[h_{\theta^{(1:H)}}(x^+)] = u_y,$$
and
$$u_h^- = \mathbb{E}_{x^-\sim\mu_{x^-}(x)}[h_{\theta^{(1:H)}}(x^-)] = \mathbb{E}_{x^-\sim\mu_{x|y^-}}[h_{\theta^{(1:H)}}(x^-)] = u_{y^-}.$$
Therefore,
$$\psi_{x,1,2}([u_h^+, u_h^-]^\top) = \psi_{x,y,y^-}(U),$$
with which
$$\mathcal{L}_{\text{cont}}^{x^+,x^-,y^-}(x, \theta) = \max_y \mathbb{E}_{y^-\sim\mu_{y^-}(x)}[\psi_{x,y,y^-}(\theta^{(H+1)}) - \psi_{x,y,y^-}(U)].$$
Since $U$ and $\theta^{(1:H)}$ do not contain $\theta^{(H+1)}$, for any $U, \bar{\theta}^{(1:H)}$, there exists $\theta^{(H+1)} = U$ for which $\psi_{x,y,y^-}(\theta^{(H+1)}) - \psi_{x,y,y^-}(U) = 0$ and hence $\mathcal{L}_{\text{cont}}^{x^+,x^-,y^-}(x, \theta) = 0$. Therefore,
$$\inf_{\theta^{(H+1)}\in\mathbb{R}^{m_y\times(m_H+1)}} \mathcal{L}_{\text{cont}}^{x^+,x^-}(x, \theta^{(1:H)}) + \mathcal{L}_{\text{cont}}^{x^+,x^-,y^-}(x, \theta^{(1:H)}, \theta^{(H+1)})$$
$$= \mathcal{L}_{\text{cont}}^{x^+,x^-}(x, \theta^{(1:H)}) + \inf_{\theta^{(H+1)}\in\mathbb{R}^{m_y\times(m_H+1)}} \mathcal{L}_{\text{cont}}^{x^+,x^-,y^-}(x, \theta^{(1:H)}, \theta^{(H+1)})$$
$$\leq \mathcal{L}_{\text{cont}}^{x^+,x^-}(x, \theta^{(1:H)}).$$

$\square$

### C.6 PROOF OF REMARK 1

*Proof of Remark 1.* For any $\theta$,

$$\mathbb{E}_{x,y}[\mathcal{L}(f_\theta(x), y)] = \inf_{\kappa \in [0,1]} \kappa \mathbb{E}_{x,y}[\mathcal{L}(f_\theta(x), y)] + (1 - \kappa)\mathbb{E}_{x,y}[\mathcal{L}(f_\theta(x), y)].$$

Applying Lemma 5 for Theorem 1 (and Lemma 7 for Theorem 6) to the second term and the assumption $\mathbb{E}_{x,y}[\mathcal{L}(f_\theta(x), y)] \leq \mathbb{E}_x[\mathcal{L}_{\mathrm{un}}(f_\theta(x))]$ to the first term yields the desired statement.

$\square$

# D  DETAILS AND DESCRIPTION OF CNGRAD

In this section we describe CNGRAD in greater detail: its implementation, different variants and run-time costs. Note that, although this section is written from the perspective of meta-tailoring, CN-GRAD is also applicable to meta-learning, we provide pseudo-code in algorithm 2. The main idea behind CNGRAD is to optimize only conditional normalization (CN) parameters $\gamma^{(l)}, \beta^{(l)}$ in the inner loop and optimize all the other weights $w$ in the outer loop. To simplify notation for implementation, in this subsection only, we overload notations to make them work over a mini-batch as follows. Let $b$ be a (mini-)batch size. Given $X \in \mathbb{R}^{b \times m_0}$, $\gamma \in \mathbb{R}^{b \times \sum_l m_l}$ and $\beta \in \mathbb{R}^{b \times \sum_l m_l}$, let $(f_{w,\gamma,\beta}(X))_i = f_{w,\gamma_i,\beta_i}(X_i)$ where $X_i$, $\gamma_i$, and $\beta_i$ are the transposes of the $i$-th row vectors of $X$, $\gamma$ and $\beta$, respectively. Similarly, $\mathcal{L}^{\text{sup}}$ and $\mathcal{L}^{\text{tailor}}$ are used over a mini-batch. We also refer to $\theta = (w, \gamma, \beta)$.

**Initialization of $\gamma, \beta$**   In the inner loop we always initialize $\gamma = \mathbf{1}_{b,\sum_l m_l}, \beta = \mathbf{0}_{b,\sum_l m_l}$. More complex methods where the initialization of these parameters is meta-trained are also possible. However, we note two things:

1. By initializing to the identity function, we can pick an architecture trained with regular inductive learning, add CN layers without changing predictions and perform tailoring. In this manner, the prediction algorithm is the same regardless of whether we trained with meta-tailoring or without the CN parameters.

2. We can add a previous normalization layer with weights $\gamma'^{(l)}, \beta'^{(l)}$ that are trained in the outer loop, having a similar effect than meta-learning an initialization. However, we do not do it in our experiments.

**First and second order versions of CNGRAD:**   $w$ affect $\mathcal{L}^{\text{sup}}$ in two ways: first, they directly affect the evaluation $f_{w,\gamma_s,\beta_s}(X)$ by being weights of the neural network; second, they affect $\nabla_\beta \mathcal{L}^{\text{tailor}}, \nabla_\gamma \mathcal{L}^{\text{tailor}}$ which affects $\gamma_s, \beta_s$ which in turn affect $\mathcal{L}^{\text{sup}}$. Similar to MAML (Finn et al., 2017), we can implement two versions: in the first order version we only take into account the first effect, while in the second order version we take into account both effects. The first order version has three advantages:

1. It is very easy to code: the optimization of the inner parameters and the outer parameters are detached and we simply need to back-propagate $\mathcal{L}^{\text{tailor}}$ with respect to $\beta, \gamma$ and $\mathcal{L}^{\text{sup}}$ with respect to $w$. This version is easier to implement than most meta-learning algorithms, since the parameters in the inner and outer loop are different.

2. It is faster: because we do not back-propagate through the optimization, the overall computation graph is smaller.

3. It is more stable to train: second-order gradients can be a bit unstable to train; this required us to lower the inner tailoring learning rate in experiments of section 5.1 for the second-order version.

The second-order version has one big advantage: it optimizes the true objective, taking into account how $\mathcal{L}^{\text{tailor}}$ will affect the update of the network. This is critical to linking the unsupervised loss to best serve the supervised loss by performing informative updates to the CN parameters.

**WarpGrad-inspired stopping of gradients and subsequent reduction in memory cost:**   Warp-Grad (Flennerhag et al., 2019) was an inspiration to CNGRAD suggesting to interleave layers that are adapted in the inner loop with layers only adapted in the outer loop. In contrast to WarpGrad, we can evaluate inputs (in meta-tailoring) or tasks (in meta-learning) in parallel, which speeds up training and inference. This also simplifies the code because we do not have to manually perform batches of tasks by iterating through them.

WarpGrad also proposes to stop the gradients between inner steps; we include this idea as an optional operation in CNGRAD, as shown in line 12 of 1. The advantage of adding it is that it decreases the memory cost when performing multiple inner steps, as we now only have to keep in memory the computation graph of the last step instead of all the steps, key when the networks are very deep like in the experiments of section 5.2. Another advantage is that it makes training more stable, reducing

variance, as back-propagating through the optimization is often very noisy for many steps. At the same time it adds bias, because it makes the greedy assumption that locally minimizing the decrease in outer loss at every step will lead to low overall loss after multiple steps.

**Computational cost:**   in CNGRAD we perform multiple forward and backward passes, compared to a single forward pass in the usual setting. In particular, if we perform $s$ tailoring steps, we execute $(s + 1)$ forward steps and $s$ backward steps, which usually take the same amount of time as the forward steps. Therefore, in its naive implementation, this method takes about $2s + 1$ times more than executing the regular network without tailoring.

However, it is well-known that we can often only adapt the higher layers of a network, while keeping the lower layers constant. Moreover, our proof about the capacity of CNGRAD to optimize a broad range of inner losses only required us to adapt the very last CN layer $\gamma^{(H)}, \beta^{(H)}$. This implies we can put the CN layers only on the top layer(s). In the case of only having one CN layer at the last network layer, we only require one initial full forward pass (as we do without tailoring). Then, we have $s$ backward-forward steps that affect only the last layer, thus costing $\frac{1}{H}$ in case of layers of equivalent cost. This leads to a factor of $1 + \frac{2s}{H}$ in cost, which for $s$ small and $H$ large (typical for deep networks), is a very small overcost. Moreover, for tailoring and meta-tailoring, we are likely to get the same performance with smaller networks, which may compensate the increase in cost.

**Meta-learning version:**   CNGRAD can also be used in meta-learning, with the advantage of being provably expressive, very efficient in terms of parameters and compute, and being able to parallelize across tasks. We show the pseudo-code for few-shot supervised learning in algorithm 2. There are two changes to handle the meta-learning setting: first, in the inner loop, instead of the unsupervised tailoring loss we optimize a supervised loss on the training (support) set. Second, we want to share the same inner parameters $\gamma, \beta$ for different samples of the same task. To do so we add the operation "repeat_interlave" (PyTorch notation), which makes $k$ contiguous copies of each parameter $\gamma, \beta$, before feeding them to the network evaluation. In doing so, gradients coming from different samples of the same task get pooled together. At test time we do the same for the $k'$ queries ($k'$ can be different than $k$). Note that, in practice, this pooling is also used in meta-tailoring when we have more than one data augmentation within $\mathcal{L}^{\text{tailor}}$.

---

**Algorithm 1:** CNGRAD for meta-tailoring

---

1 **Subroutine** *Training($f$, $\mathcal{L}^{\text{sup}}$, $\lambda_{sup}$, $\mathcal{L}^{\text{tailor}}$, $\lambda_{tailor}$, $steps$, $((x_i, y_i))_{i=1}^{n}$)*

2     randomly initialize $w$  // All parameters except $\gamma, \beta$; trained in outer loop

3     **while** *not done* **do**

4         **for** $0 \leq i \leq n/b$ **do**                       // $b$ batch size

5             $X, Y = x_{ib:i(b+1)}, y_{ib:i(b+1)}$

6             $\gamma_0 = \mathbf{1}_{b, \sum_l m_l}$

7             $\beta_0 = \mathbf{0}_{b, \sum_l m_l}$

8             **for** $1 \leq s \leq steps$ **do**

9                 $\gamma_s = \gamma_{s-1} - \lambda_{tailor} \nabla_\gamma \mathcal{L}^{\text{tailor}}(w, \gamma_{s-1}, \beta_{s-1}, X)$

10                 $\beta_s = \beta_{s-1} - \lambda_{tailor} \nabla_\beta \mathcal{L}^{\text{tailor}}(w, \gamma_{s-1}, \beta_{s-1}, X)$

11                 $w = w - \lambda_{sup} \nabla_w \mathcal{L}^{\text{sup}}\left(f_{w, \gamma_s, \beta_s}(X), Y\right)$

12                 $\beta_s, \gamma_s = \beta_s.detach(), \gamma_s.detach()$     // Optional operation: WarpGrad detach to avoid back-proping through multiple steps; reducing memory, and increasing stability, but adding bias.

13     **return** $w$

14 **Subroutine** *Prediction($f$, $w$, $\mathcal{L}^{\text{tailor}}$, $\lambda$, $steps$, $X$)*     // For meta-tailoring & tailoring

        // X contains multiple inputs, with independent tailoring processes

15     $b = X.shape[0]$                      // number of inputs

16     $\gamma_0 = \mathbf{1}_{b, \sum_l m_l}$

17     $\beta_0 = \mathbf{0}_{b, \sum_l m_l}$

18     **for** $1 \leq s \leq steps$ **do**

19         $\gamma_s = \gamma_{s-1} - \lambda \nabla_\gamma \mathcal{L}^{\text{tailor}}(w, \gamma_{s-1}, \beta_{s-1}, X)$

20         $\beta_s = \beta_{s-1} - \lambda \nabla_\beta \mathcal{L}^{\text{tailor}}(w, \gamma_{s-1}, \beta_{s-1}, X)$

21     **return** $f_{w, \gamma_{steps}, \beta_{steps}}(X)$

---

**Algorithm 2:** CNGRAD for meta-learning

1 **Subroutine** *Meta-training(f, $\mathcal{L}^{\sup}$, $\lambda_{inner}$, $\lambda_{outer}$, steps,$\mathcal{T}$)*
2   randomly initialize $w$  // All parameters except $\gamma,\beta$; trained in outer loop
3   **while** *not done* **do**
4     **for** $0 \le i \le n/b$ **do**                               // $b$ batch size
5       $X_{train}, Y_{train} = [\,], [\,]$
6       $X_{test}, Y_{test} = [\,], [\,]$
7       **for** $ib \le j \le i(b+1)$ **do**
8         $(inp, out) \sim_k \mathcal{T}_j$     // Take $k$ samples from each task for training
9         $X.append\,(inp)\,;Y.append\,(out)$
10         $(query, target) \sim'_k \mathcal{T}_j$  // Take $k'$ samples from each task for testing
11         $X.append\,(query)\,;Y.append\,(target)$
               // We can now batch evaluations of multiple tasks
      $X_{train}, Y_{train} = concat\,(X_{train}, dim = 0)\,, concat\,(Y_{train}, dim = 0)$
12       $X_{test}, Y_{test} = concat\,(X_{test}, dim = 0)\,, concat\,(Y_{test}, dim = 0)$
13       $\gamma_0 = \mathbf{1}_{b, \sum_l m_l}$
14       $\beta_0 = \mathbf{0}_{b, \sum_l m_l}$
15       **for** $1 \le s \le steps$ **do**
               // We now repeat the CN parameters $k$ times so that samples from the same task share the same CN parameters
16         $\gamma^{tr}_{s-1}, \beta^{tr}_{s-1} = \gamma_{s-1}.repeat\_interleave(k, 1), \beta_{s-1}.repeat\_interleave(k, 1)$
17         $\gamma_s = \gamma_{s-1} - \lambda_{innner}\nabla_\gamma \mathcal{L}^{\sup}(f_{w, \gamma^{tr}_{s-1}, \beta^{tr}_{s-1}}(X_{train}), Y_{train})$
18         $\beta_s = \beta_{s-1} - \lambda_{innner}\nabla_\beta \mathcal{L}^{\sup}(f_{w, \gamma^{tr}_{s-1}, \beta^{tr}_{s-1}}(X_{train}), Y_{train})$
19         $\gamma^{test}_s, \beta^{test}_s = \gamma_s.repeat\_interleave(k', 1), \beta_s.repeat\_interleave(k', 1)$
20         $w = w - \lambda_{outer}\nabla_w \mathcal{L}^{\sup}\left(f_{w, \gamma^{test}_s, \beta^{test}_s}(X_{test}), Y_{test})\right)$
21         $\beta_s, \gamma_s = \beta_s.detach(), \gamma_s.detach()$
               // WarpGrad detach to not backprop through multiple steps
22   **return** $w$
23 **Subroutine** *Meta-test(f, w, $\mathcal{L}^{\sup}$, $\lambda_{inner}$,steps,$X_{train}$, $Y_{train}$, $X_{test}$)*
    // Assuming a single task, although we could evaluate multiple tasks in parallel as in meta-training.
24   $\gamma_0 = \mathbf{1}_{1, \sum_l m_l}$         // single $\gamma,\beta$ because we only have one task
25   $\beta_0 = \mathbf{0}_{1, \sum_l m_l}$
26   **for** $1 \le s \le steps$ **do**
27     $\gamma^{tr}_{s-1}, \beta^{tr}_{s-1} = \gamma_{s-1}.repeat\_interleave(k, 1), \beta_{s-1}.repeat\_interleave(k, 1)$
28     $\gamma_s = \gamma_{s-1} - \lambda_{innner}\nabla_\gamma \mathcal{L}^{\sup}(f_{w, \gamma^{tr}_{s-1}, \beta^{tr}_{s-1}}(X_{train}), Y_{train})$
29     $\beta_s = \beta_{s-1} - \lambda_{innner}\nabla_\beta \mathcal{L}^{\sup}(f_{w, \gamma^{tr}_{s-1}, \beta^{tr}_{s-1}}(X_{train}), Y_{train})$
30   $\gamma^{test}_{steps}, \beta^{test}_{steps} = \gamma_{steps}.repeat\_interleave(k', 1), \beta_{steps}.repeat\_interleave(k', 1)$
31   **return** $f_{w, \gamma^{test}_{steps}, \beta^{test}_{steps}}(X_{test})$

## E    Experimental details of physics experiments

**Dataset generation**    As mentioned in the main text, 5-body systems are chaotic and most random configurations are unstable. To generate our dataset we used Finite Differences to optimize 5-body dynamical systems that were stable for 200 steps (no planet collisions and no planet outside a predetermined grid) and then picked the first 100 steps of their trajectories, to ensure dynamical stability. To generate each trajectory, we randomly initialized 5 planets within a 2D grid of size $w = 600, h = 300$, with a uniform probability of being anywhere in the central grid of size $w/2, h/2$, each with a mass sampled from a uniform between $[0.15, 0.25]$ (arbitrary units) and with random starting velocity initialized with a Gaussian distribution. We then use a 4th order Runge-Kutta integrator to accurately simulate the ODE of the dynamical system until we either reach 200 steps, two planets get within a certain critical distance from each other or a planet gets outside the pre-configured grid. If the trajectory reached 200 steps, we added it to the dataset; otherwise we made a small random perturbation to the initial configuration of the planets and tried again. If the new perturbation did not reach 200 steps, but lasted longer we kept the perturbation as the new origin for future initialization perturbations, otherwise we kept our current initialization. Once all the datasets were generated we picked those below a threshold mean mass and partitioned them randomly into train and test. Finally, we normalize each of the 25 dimensions (5 planets and for each planet $x, y, v_x, v_y, m$) to have mean zero and standard deviation one. For inputs, we use each state and as target we use the next state; therefore, each trajectory gives us 100 pairs.

For more details, we attach the code that generated the dataset.

**Implementation of tailoring, meta-tailoring and** CNGRAD    All of our code is implemented in PyTorch (Paszke et al., 2019), using the higher library (Grefenstette et al., 2019)(`https://github.com/facebookresearch/higher`) to implement the second-order version of CNGRAD. We implemented a 3-layer feedforward neural network, with a conditional normalization layer after each layer except the final regression layer. The result of the network was added to the input, thus effectively predicting the delta between the current state and the next state. For both the first-order and second-order versions of CNGRAD, we used the detachment of WarpGrad (line 12 in algorithm 1). For more details, we also attach the implementation of the method.

**Compute and hyper-parameter search**    To keep the evaluation as strict as possible, we searched all the hyper-parameters affecting the inductive baseline and our tailoring versions with the baseline and simply copied these values for tailoring and meta-tailoring. For the latter two, we also had to search for $\lambda_{tailor}$.

The number of epochs was 1000, selected with the inductive baseline, although more epochs did not substantially affect performance in either direction. We note that meta-tailoring performance plateaued earlier in terms of epochs, but we left it the same for consistency. Interestingly, we found that regularizing the physics loss (energy and momentum conservation) helped the inductive baseline, even though the training data already has 0 physics loss. We searched over $[10^{-4}, 3 \cdot 10^{-4}, 10^{-3}, 3 \cdot 10^{-3}, 10^{-2}]$ for the weight assigned to the physics loss and chose $2 \cdot 10^{-3}$ for best performance in the inductive baseline. To balance between energy and momentum losses we multiplied the momentum loss by 10 to roughly balance their magnitudes before adding them into a single physics loss, this weighting was not searched. We copied these settings for meta-tailoring.

In terms of the neural network architecture, we chose a simple model with 3 hidden layers of the same size and tried $[128, 256, 512]$ on the inductive baseline, choosing 512 and deciding not to go higher for compute reasons and because we were already able to get much lower training loss than test loss. We copied these settings for the meta-tailoring setup. We note that since there are approximately $O(m_h^2)$ weight parameters, yet only $O(m_h)$ affine parameters used for tailoring, adding tailoring and meta-tailoring increase parameters roughly by a fraction $O(1/m_h)$, or about $0.2\%$. Also in the inductive baseline, we tried adding Batch Normalization (Ioffe & Szegedy, 2015), since it didn't affect performance we decided not to add it.

We chose the tailoring step size parameter by trying $[10^{-5}, 10^{-4}, 10^{-3}, 10^{-2}]$, finding $10^{-3}$ worked well while requiring less steps than using a smaller step size. We therefore used this step for meta-tailoring as well, which worked well for first-order CNGRAD, but not for second-order CNGRAD, whose training diverged. We thus lowered the tailoring step size to $10^{-4}$ for the second-order version,

which worked well. We also tried clipping the inner gradients to increase the stability of the second-order method; since gains on preliminary experiments were small, we decided to keep it out for simplicity.

For meta-tailoring we only tried 2 and 5 tailoring steps (we wanted more than one step to show the algorithm capability, but few tailoring steps to keep inference time competitive). Since they worked similarly well, we chose 2 steps to have a faster model. For the second-order version we also used 2 steps, which performed much better than the inductive baseline and tailoring, but worse than the first-order version reported in the main text(about $20\%$ improvement of the second-order version vs. $7\%$ of tailoring and $35\%$ improvement of the first-order version).

For the baseline of optimizing the output we tried a step size of $10^{-4}, 10^{-3}, 10^{-2}, 10^{-1}$. Except for a step size of $10^{-1}$, results optimized the physics loss and always achieved a very small improvement, without overfitting to the physics loss. We thus chose a big learning rate and high number of steps to report the biggest improvement, of $0.7\%$.

**Runs, compute and statistical confidence:**   we ran each method 2 times and averaged the results. Note that the baseline of optimizing the output and *tailoring* start from the inductive learning baseline, as they are meant to be methods executed after regular inductive training. This is why both curves start at the same point in figure 2. For those methods, we report the standard deviation of the mean estimate of the *improvement*, since they are executed on the same run. Note that the standard deviation of the runs would be higher, but less appropriate. For meta-tailoring, we do use the standard deviation of the mean estimate of both runs, since they are independent from the inductive baseline.

All experiments were performed on a GTX 2080 Ti with 4 CPU cores.

## F   UNDERSTANDING THE DIFFERENCE BETWEEN TAILORING AND TEST-TIME TRAINING WITH EXPERIMENTS ON THE PLANETS DATASET

As discussed in section 2, multiple methods have been proposed to optimize unsupervised objectives at test time to account for some deficiency in the training distribution. This makes a lot of sense: it has been widely observed that training a deep model in some task can learn useful features so that fine-tuning it on small amounts of data for a different task achieves high performance (Donahue et al., 2014). When having access to some unlabeled test data that differs from the training data in some way (distribution, task, image scale, simulation vs real, etc) we would want to fine-tune the model to account for this, but by definition, we do not have labels. Therefore, multiple methods have been proposed to optimize an *unsupervised* loss to adapt to the test.

Probably the closest to *tailoring* (but not meta-tailoring) is "Test-time training with self-supervision for out-of-distribution generalization" (Sun et al., 2019) which uses available test samples to approximate the test distribution when there is a distribution shift and optimizes an unsupervised loss to adapt to that distribution. There are three key differences with our work.

1. In contrast to test-time-training for out-of-distribution generalization and other prior work, we suggest that adaptation is useful even when there is no change between training and test distribution or task, solving different problems. We show test-time training is not good for same-distribution generalization and that meta-tailoring is not the right approach for out-of-distribution generalization.

2. Because adaptation makes sense even on the same distribution, it also makes sense to do tailoring at training time, not just at test time, resulting in meta-tailoring. Moreover, this work shows theoretically and empirically shown that meta-tailoring is a better paradigm than simply tailoring because, when training and test come from the same task and distribution, we want to train as we test.[1]

3. Test-time training and other prior work suggest to adapt via unsupervised samples on available data from the correct test distribution. This suggest two testable consequences that test-time training predicts but tailoring doesn't; we show neither is satisfied when the distributions are the same:

   (a) Since samples are used to adapt to the test distribution, most of the benefit of adapting with sample $x$ before evaluating on $x$ should be captured by adapting on another *test* sample $x' \neq x$. We show this is not the case when training matches test.

   (b) As we get more test samples performance should increase (and it indeed does in their OOD experiments). We show this is not the case when the distributions are the same: even with 6400 samples (a full batch, including the evaluated sample), performance is worse than just tailoring on the one trained sample. Meta-tailoring further increases this gap.

**Related work on unsupervised adaptation are solving different problems**    "Test-time training with self-supervised objectives for out-of-distribution generalization"(Sun et al., 2019) focuses on problems where there is a distribution shift between training and testing. Other methods have used similar unsupervised fine-tuning methods to adapt from simulated to real data or to different tasks in the meta-learning context. The key insight of this work (which echoes that of Vapnik in the 80s) is that we can tailor the model to each query (both test and training queries) at prediction-time.

The single-sample case of test-time training is computationally equivalent to tailoring when the adaptation sample is the same as the evaluated sample (which is the most common case). However, this new application of fine-tuning motivates meta-tailoring, which also applies it at training time in an inner loop. At the end of this section we show experimental evidence that test-time training is worse than tailoring in the classic ML setting. Conversely, meta-tailoring (which is well-motivated for same-distribution generalization) may not be well-suited for extrapolation: first, our theoretical guarantees no longer apply and second, we experimentally tested on an extrapolation version of the planets dataset (asking to generalize to bigger planets) and meta-tailoring performs worse than just

---

[1]We call our paper *tailoring* instead of *meta-tailoring* because the key idea is that fine-tuning is useful to encode inductive biases in the classic learning paradigm where test matches training. Moreover, meta-tailoring could confuse readers into believing we are in a meta-learning paradigm.

tailoring. The intuition for why this may be the case is that meta-tailoring couples the unsupervised loss to the task loss; however, this coupling is learned for the training distribution, not the test distribution. When applied to a different distribution, this coupling may be incorrect.

**Meta-tailoring is the correct paradigm for classic generalization**  Our theoretical guarantees in section 3 apply to meta-tailoring, but not tailoring. This is because in machine learning to get guarantees we usually need to test as we train; which meta-tailoring does, but tailoring and test-time training do not. Note that the guarantees done by test-time training (Sun et al., 2019), which would also apply to tailoring, assume a relation between the two gradients when the model is already trained. This *a posteriori* assumption is much more strict than the *a priori* assumptions we use, which do not need to make assumptions about the *trained* model and instead guarantee that (meta-)training will happen in such a way that the conditions will hold for the trained model.

Finally, all our experiments use meta-tailoring to obtain results, since it also performs better in practice.

**Experiments show that tailoring is a better approach than test-time training in same-distribution generalization.**  Test-time training (and other fine-tuning for adaptation methods) suggest to fine-tune on the available samples. The most useful case is when we use the single test sample for itself (tailoring). However, we can make the artificial experiment of using a different sample for adaptation than the one we test. This separates the *adaptation* component from the *tailoring* component. Our experiments showed that optimization on a different sample is much slower (50 steps vs less than 10) and results in less than half the improvement than tailoring ($2.9\%$ vs $7.5\%$). Notably, meta-tailoring adapts even faster than tailoring, and to much larger improvements ($36\%$).

Possibly more surprising, we also experiment on allowing test-time training to adapt on the entire test batch, which by definition will now include each sample we evaluate on. Each batch contained 64 trajectories of 100 states, thus 6400 samples, 1 of which is the test query and 99 others are from the same trajectory, and thus specially related. Note that this is not allowed in the classic test evaluation where test samples do not see each other.

For test-time training and other fine-tuning methods, having 6400 times more data would be extremely useful to have much better adaptation. From the tailoring point-of-view, however, the extra samples are a nuisance, since it is now harder to tailor the model to each individual sample. We experimentally find that 6400-sample test-time training slightly improves over single-sample test-time training, but is still much worse than tailoring and meta-tailoring: $3.6\%$ vs $7.5\%$, $36\%$ improvement. Note that, since the evaluated query is part of the 6400 optimized samples, this provides strong evidence that the other 6399 samples are hurting the fine-tuning, aligning with the tailoring approach instead of the adaptation approach.

## G  TOY ADVERSARIAL EXAMPLES EXPERIMENT

We illustrate the tailoring process with a simple illuminating example using the data from Ilyas et al. (2019). They use discriminant analysis on the data in figure 5 and obtain the purple linear separator. It has the property that, under assumptions about Gaussian distribution of the data, points above the line are more likely to have come from the blue class, and those below, from the red class. This separator is not very adversarially robust, in the sense that, for many points, a perturbation with a small $\delta$ would change the assigned class. We improve the robustness of this classifier by tailoring it using the loss $\mathcal{L}^{\text{tailor}}(x, \theta) = \text{KL}(\phi(f_\theta(x)) \mid\mid \phi(f_\theta(x + \arg\max_{|\delta| < \varepsilon} \sum_j e^{f_\theta(x+\delta)_j})))$, where KL represents the KL divergence, $\phi$ is the logistic function, and $\phi(f_\theta(x)_i)$ is the probability of $x$ being in class $i$, so that $\phi(f_\theta(x))$ represents the entire class distribution.

With this loss, we can adjust our parameters $\theta$ so that the KL divergence between our prediction at $x$ is closer to the prediction at perturbed point $x + \delta$, over all perturbations in radius $\varepsilon$. Each of the curves in figure 3 corresponds to a decision boundary induced by tailoring the original separator with a different value for the maximum perturbation $\varepsilon$. Note that the resulting separators are non-linear, even though we are tailoring a linear separator, because the tailoring is local to the prediction point. We also have the advantage of being able to choose different values of $\varepsilon$ at prediction time.

**Hyper-parameters**  the model does not have any hyper-parameters, as we use the model from Ilyas et al. (2019), which is based on the mean $\mu$ and standard deviation $\sigma$ of the Gaussians. For tailoring, we used a $5 \times 5$ grid to initialize the inner optimization to find the point of highest probability within the $\epsilon$-ball. Using a single starting point did not work as gradient descent found a local optima. Using more $(10 \times 10)$ did not improve results further, while increasing compute. We also experimented between doing a weighted average of the predictions by their energy to compute the tailoring loss or picking the element with the biggest energy. Results did not seem to differ much (likely because likelihood distributions are very peaked), so we picked the simplest option of imitating the element of highest probability. Doing a single tailoring step already worked well (we tried step sizes of $10^{-1}, 1, 10, 30$ with 10 working best), so we kept that for simplicity and faster predictions.

Regarding compute, this experiment can be generated in a few minutes using a single GTX 2080 Ti.

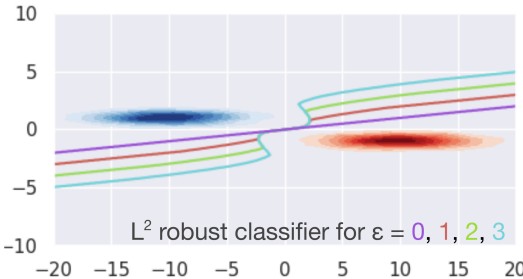

Figure 5: Decision boundary of our model at multiple levels of robustness on an example from Ilyas et al. (2019).

| $\sigma$ | Method | 0.0 | 0.25 | 0.5 | 0.75 | 1.0 | 1.25 | 1.5 | 1.75 | 2.00 | 2.25 | ACR |
|---|---|---|---|---|---|---|---|---|---|---|---|---|
| 0.25 | RandSmooth | 0.75 | 0.60 | 0.43 | 0.26 | 0.00 | 0.00 | 0.00 | 0.00 | 0.00 | 0.00 | 0.416 |
| | Meta-tailored | **0.80** | **0.66** | **0.48** | 0.29 | 0.00 | 0.00 | 0.00 | 0.00 | 0.00 | 0.00 | **0.452** |
| 0.50 | RandSmooth | 0.65 | 0.54 | 0.41 | 0.32 | 0.23 | 0.15 | 0.09 | 0.04 | 0.00 | 0.00 | 0.491 |
| | Meta-tailored | 0.68 | 0.57 | 0.45 | **0.33** | 0.23 | 0.15 | 0.08 | 0.04 | 0.00 | 0.00 | **0.542** |
| 1.00 | RandSmooth | 0.47 | 0.39 | 0.34 | 0.28 | 0.21 | 0.17 | **0.14** | 0.08 | 0.05 | 0.03 | 0.458 |
| | Meta-tailored | 0.50 | 0.43 | 0.36 | 0.30 | **0.24** | **0.19** | **0.14** | **0.10** | **0.07** | **0.05** | **0.546** |

Figure 6: Percentage of points with certificate above different radii, and average certified radius (ACR) for on the CIFAR-10 dataset. Meta-tailoring improves the Average Certification Radius by $8.6\%, 10.4\%, 19.2\%$ respectively. Results for Cohen et al. (2019) are taken from Zhai et al. (2020) because they add more measures than the original work, with similar results.

| $\sigma$ | Method | 0.0 | 0.25 | 0.5 | 0.75 | 1.0 | 1.25 | 1.5 | 1.75 | 2.00 | 2.25 | ACR |
|---|---|---|---|---|---|---|---|---|---|---|---|---|
| 0.25 | RandSmooth | 0.75 | 0.60 | 0.43 | 0.26 | 0.00 | 0.00 | 0.00 | 0.00 | 0.00 | 0.00 | 0.416 |
| | Salman | 0.74 | 0.67 | 0.57 | 0.47 | 0.00 | 0.00 | 0.00 | 0.00 | 0.00 | 0.00 | 0.538 |
| | MACER | **0.81** | **0.71** | **0.59** | 0.43 | 0.00 | 0.00 | 0.00 | 0.00 | 0.00 | 0.00 | **0.556** |
| | Meta-tailored | 0.80 | 0.66 | 0.48 | 0.29 | 0.00 | 0.00 | 0.00 | 0.00 | 0.00 | 0.00 | 0.452 |
| 0.50 | RandSmooth | 0.65 | 0.54 | 0.41 | 0.32 | 0.23 | 0.15 | 0.09 | 0.04 | 0.00 | 0.00 | 0.491 |
| | Salman | 0.50 | 0.46 | 0.44 | 0.40 | 0.38 | 0.33 | 0.29 | 0.23 | 0.00 | 0.00 | 0.709 |
| | MACER | 0.66 | 0.60 | 0.53 | **0.46** | 0.38 | 0.29 | 0.19 | 0.12 | 0.00 | 0.00 | **0.726** |
| | Meta-tailored | 0.68 | 0.57 | 0.45 | 0.33 | 0.23 | 0.15 | 0.08 | 0.04 | 0.00 | 0.00 | 0.542 |
| 1.00 | RandSmooth | 0.47 | 0.39 | 0.34 | 0.28 | 0.21 | 0.17 | 0.14 | 0.08 | 0.05 | 0.03 | 0.458 |
| | Salman | 0.45 | 0.41 | 0.38 | 0.35 | **0.32** | 0.28 | **0.25** | **0.22** | **0.19** | **0.17** | 0.787 |
| | MACER | 0.45 | 0.41 | 0.38 | 0.35 | **0.32** | 0.29 | **0.25** | **0.22** | 0.18 | 0.16 | **0.792** |
| | Meta-tailored | 0.50 | 0.43 | 0.36 | 0.30 | 0.24 | 0.19 | 0.14 | 0.10 | 0.07 | 0.05 | 0.546 |

Figure 7: Percentage of points with certificate above different radii, and average certified radius (ACR) for on the CIFAR-10 dataset, comparing with SOA methods. In contrast to pretty competitive results in ImageNet, meta-tailoring improves randomized smoothing, but not enough to reach SOA. It is worth noting that the SOA algorithms could also likely be improved via meta-tailoring.

## H  EXPERIMENTAL DETAILS OF ADVERSARIAL EXPERIMENTS

Results for CIFAR-10 experiments can be found in table 6, 7; results for ImageNet comparing to state-of-the-art methods can be found in 8.

**Hyper-parameters and other details of the experiments**  there are just three hyper-parameters to tweak for these experiments, as we try to remain as close as possible to the experiments from Cohen et al. (2019). In particular, we tried different added noises $\nu \in [0.05, 0.1, 0.2]$ and tailoring inner steps $\lambda \in [10^{-3}, 10^{-2}, 10^{-1}, 10^0]$ for $\sigma = 0.5$. To minimize compute, we tried these settings by tailoring (not meta-tailoring) the original model and seeing its effects on the smoothness and stability of optimization, choosing $\nu = 0.1, \lambda = 0.1$ (the fact that they're the same is a coincidence). We chose to only do a single tailoring step to reduce the computational burden, since robustness certification is very expensive, as each example requires 100k evaluations (see below). For simplicity and to avoid excessive tuning, we chose the hyper-parameters for $\sigma = 0.5$ and copied them for $\sigma = 0.25$ and $\sigma = 1$. As mentioned in the main text, $\sigma = 1$ required initializing our model with that of Cohen et al. (2019) (training wasn't stable otherwise), which is easy to do using CNGRAD.

In terms of implementation, we use the codebase of Cohen et al. (2019)(`https://github.com/locuslab/smoothing`) extensively, modifying it only in a few places, most notably in the architecture to include tailoring in its forward method. It is also worth noting that we had to deactivate their disabling of gradients during certification, because tailoring requires gradients. We chose to use the first-order version of CNGRAD which made it much easier to keep our implementation very close to the original. It is likely that doing more tailoring steps would result in better performance.

| $\sigma$ | Method | 0.0 | 0.5 | 1.0 | 1.5 | 2.0 | 2.5 | 3.0 | ACR |
|---|---|---|---|---|---|---|---|---|---|
| 0.25 | RandSmooth | 0.67 | 0.49 | 0.00 | 0.00 | 0.00 | 0.00 | 0.00 | 0.470 |
| | Salman | 0.65 | 0.56 | 0.00 | 0.00 | 0.00 | 0.00 | 0.00 | 0.528 |
| | MACER | 0.68 | **0.57** | 0.00 | 0.00 | 0.00 | 0.00 | 0.00 | **0.544** |
| | Meta-tailored RS | **0.72** | 0.55 | 0.00 | 0.00 | 0.00 | 0.00 | 0.00 | 0.494 |
| 0.50 | RandSmooth | 0.57 | 0.46 | 0.37 | 0.29 | 0.00 | 0.00 | 0.00 | 0.720 |
| | Salman | 0.54 | 0.49 | **0.43** | **0.37** | 0.00 | 0.00 | 0.00 | 0.815 |
| | MACER | 0.64 | 0.53 | **0.43** | 0.31 | 0.00 | 0.00 | 0.00 | **0.831** |
| | Meta-tailored RS | 0.66 | 0.54 | 0.42 | 0.31 | 0.00 | 0.00 | 0.00 | 0.819 |
| 1.00 | RandSmooth | 0.44 | 0.38 | 0.33 | 0.26 | 0.19 | 0.15 | 0.12 | 0.863 |
| | Salman | 0.40 | 0.38 | 0.33 | 0.30 | **0.27** | **0.25** | **0.20** | 1.003 |
| | MACER | 0.48 | 0.37 | 0.34 | 0.30 | 0.25 | 0.18 | 0.14 | 1.008 |
| | Meta-tailored RS | 0.52 | 0.45 | 0.36 | 0.31 | 0.24 | 0.20 | 0.15 | **1.032** |

Figure 8: Percentage of points with certificate above different radii, and average certified radius (ACR) for on the ImageNet dataset, including other SOA methods. Randomized smoothing with meta-tailoring are very competitive with other SOA methods, including having the biggest ACR for $\sigma = 1$.

We note that other works focused on adversarial examples, such as Zhai et al. (2020); Salman et al. (2019), improve on Cohen et al. (2019) by bigger margins. However, tailoring and meta-tailoring can also improve a broad range of algorithms in applications outside of adversarial examples. Moreover, they could also improve these new algorithms further, as these algorithms can also be tailored and meta-tailored.

**Compute requirements**   For the CIFAR-10 experiments building on Cohen et al. (2019), each training of the meta-tailored method was done in a single GTX 2080 Ti for 6 hours. Certification was much more expensive (10k examples with 100k predictions each for a total of $10^9$ predictions). Since certifications of different images can be done in parallel, we used a cluster consisting of 8 GTX 2080 Ti, 16 Tesla V-100, and 40 K80s (which are about 5 times slower), during 36 hours.

For the ImageNet experiments, we fine-tuned the original models for 5 epochs; each took 18 hours on 1 Tesla V-100. We then used 30 Tesla V-100 for 20 hours for certification.

# I  EXPERIMENTAL DETAILS OF TAILORING FOR MODEL-BASED REINFORCEMENT LEARNING

We re-implemented PDDM (Nagabandi et al., 2020) in Pytorch (Paszke et al., 2019). We keep all of its hyper-parameters the same between the base inductive version and the meta-tailoring versions. We note that, in the process, we found out that some of the hyper-parameters for the dclaw environemnt were wrongly set in their github and instead were correctly specified in their appendix. We could not reproduce the other complex environment, baoding, and are currently in conversations with them trying to reproduce it. This (current) lack of reproducibility has been confirmed by the authors as well as Github issues raised by other researchers.

There are three main hyper-parameters we tried changing: the learning rate of the likelihood model (we tried $10^{-2}, 10^{-3}, 10^{-4}$, and settled for $10^{-3}$), the number of Gaussians (we set it to $5$ without parameter search), and the tailoring learning rate: we tried $10^{-2}, 10^{-3}, 10^{-4}, 10^{-5}$ and show $10^{-3}$ as large tailoring and $10^{-4}$ as small. $10^{-2}$ was too unstable to obtain proper results.

A few implementation tricks were required to make training stable:

- We clamped the affine parameters after tailoring between $-10$ and $10$.
- We set a minimum predicted standard deviation of each Gaussian in the mixture of Gaussians to $10^{-2}$, to avoid degenerate Gaussians.

