# OpenReview forum: "Tailoring: encoding inductive biases by optimizing unsupervised objectives at prediction time"
_ICLR.cc/2021/Conference — Reject_

### Official Review · AnonReviewer4 · 2020-10-27
**Tailoring - An interesting approach for more powerful inductive biases**

**Rating:** 7
**Confidence:** 3

**Review:**

The paper proposes tailoring and meta-tailoring, learning processes that fine-tune the model parameters during test-time using unsupervised objectives. This allows for designing and integrating powerful inductive biases into the model, leading to an improved test-time performance in two example tasks.

Strengths:
* The proposed approaches are well-motivated, and the paper is written clearly.
* CNGrad provides an elegant solution to implement tailoring efficiently.
* The experimental results show the benefits of the proposed approach convincingly.

Weaknesses:
* While CNGrad provides an efficient implementation of tailoring, it would be interesting to see how it compares to its “inefficient counterpart”, in which no additional parameters are introduced, but the parameters w are optimized for each individual sample using the supervised and the tailoring objective simultaneously.

* It would be nice to include the inductively trained model as a baseline to the experiment in section 5.2. This could highlight more clearly the benefit of tailoring when applied to adversarial examples.

* I find the statement that the paper showed “the applicability of tailoring on three domains” slightly misleading. The paper shows its applicability experimentally in two domains, and shows theoretical results for the third. Additionally, building on these theoretical results, it would be interesting to see how the contrastive loss might be used for tailoring in an experimental setting.

Questions:
* How is the element-wise normalization in CNGrad performed at test-time? Do you keep a running average of the batch-statistics as is done in batch normalization?

* Which loss was used for the results in Table 1?

* What are the run-time implications of tailoring, both at train and test time?

Additional Comments:
* Fig 1: Try to avoid using red and green as distinguishing colors to improve the paper’s accessibility.

* It might be nice to add a paragraph on inductive learning to the intro.


Disclaimer: I did not check the provided proofs in detail.

---

> ### Author Response · Authors · 2020-11-24
> **Details on runtime and inductive learning paradigm, more results in general answer**
>
> Thanks for your review. We address your points in order, and note that the answer to all reviewers includes general comments on recurrent topics as well as new experiments.
>
> **Inefficient meta-tailoring vs CNGrad** Compute-wise the naive implementation of meta-tailoring (using MAML or similar algorithms) is significantly slower than CNGrad by a factor between 128 (batch for ImageNet) and 6400 (batch for planet experiments), making it far too slow to run (on the order of weeks to months). This is because it has to be run sequentially and the affine layers are very cheap computationally compared to linear layers. Result-wise, our theorem in section 4 guarantees that we have enough capacity to optimize the inner loop with the affine parameters alone. However, CNGrad still needs to optimize all parameters in the outer loop.
>
> **Classical inductive baseline & setting** Because inductive learning is classic machine learning (train model, freeze parameters, test), section 5.2 is comparing vanilla (inductive) RandomizedSmoothing vs. its meta-tailoring counterpart. We have added “Inductive” in the table to make it clearer. This is also relevant to one of your comments, suggesting we add a paragraph on inductive learning in the intro. The paragraph was there (starting with “In classic supervised learning“), but also didn’t mention inductive learning. We have changed it to start with“In classic inductive supervised learning” to make it clear.
>
> **Contrastive experiments & theoretical guarantees** In the general response to all reviewers, we address the confusion regarding the contrastive experiments and add more details on the theoretical guarantees. There, we also discuss new results on model-based RL.
>
> **BatchNorm statistics** “Do you keep a running average of the batch-statistics for CNGrad at test time?” CNGrad per se does not apply any normalization, only an affine transformation. Therefore, there is no need to keep any statistics. One can combine BatchNorm and CNGrad, handling the batch statistics of BatchNorm as per usual.
> Table 1 used MSE loss; we’ve added that to the description.
>
> **“Run-time implications of tailoring both at training and test-time”**
> Good question. The detailed answer can be found on “Computational cost” of appendix D. In short, first, it depends on the number of tailoring steps linearly (often 1 is enough). Then, it depends on which layers are adapted in the inner loop: if we add affine transformations to all the layers in the network and do one tailoring step, performance will be 3x the usual prediction time (initial forward pass, tailoring adaptation, final forward pass). One can speed this up (which we did in the adversarial experiments) by only adapting the higher layers (which are usually the ones needing adaptation). This reduces the computational factor to $1+\frac{2a}{L}$, where ‘a’ is the number of adapted layers and L the number of total layers. For instance, if we tailor the last 5 layers of a resnet-110 then we would get an increase only of $5\cdot2/110 \approx 9$ percent, (1.09x factor).
>
> **Color-blind-friendly plots** Good point on the colors for the plots, we have changed the red to light orange and made the green darker. We also checked with a color-blind simulator that they are distinguishable. Thanks for the suggestion.

---

### Official Review · AnonReviewer2 · 2020-10-28
**A very interesting idea, but needs more empirical validation of its claims.**

**Rating:** 5
**Confidence:** 2

**Review:**

The authors propose a learning method called tailoring, inspired
by transductive learning, which works
by fine tuning a model on an unsupervised loss given a test time example
input. The benefit of this approach is that arbitrary constraints
on predictions can be imposed without a suffering from a generalization gap
if the constraints were used as an auxiliary objective during training.
The authors also propose meta-tailor, which includes the tailoring process
as an inner optimization loop during training. In a sense, meta-tailoring
is like meta-learning but with  each training example considered a separate
task.

The authors provided extensive theoretical justification for tailoring
and meta tailoring, as well as an efficient means to implement it through
conditional normalization parameters.
They then perform two experiments, one where (meta-) tailoring
is used to impose physical constraints on a neural network physics simulator
and another where meta-tailoring is used to improve the robustness of an
image classifier to adversarial attacks.


The idea of (meta-)tailoring is quite intriguing and I could see it
applied to scenarios in structured prediction or as an alternative to
posterior regularization. However, this paper was very theory heavy and I must admit,
I struggled to understand the import or necessity of the provided theorems.
One of the claims of this paper is that they demonstrate that improving
prediction quality with contrastive learning, but this is only done
theoretically. As this claim is included in the list of experimental results,
I find that somewhat disingenuous. I would have much preferred a
contrastive learning experiment.

I lean to reject this paper.



Miscellaneous note:

In the last paragraph of page 4, it is mentioned that parts of
definition 1 and theorem 1 are in bold green, but I see no bold green in
this copy of the paper.

---

> ### Author Response · Authors · 2020-11-24
> **More details on theory and pointing to general response for new experiments**
>
> Thank you for your comments.
>
> “I struggled to understand the importance or necessity of the provided theorems” & “I see no bold green in theorem 1 and definition 1”
>
> **The theory section formalizes and provides guarantees to the claims and intuitions made in the introduction.** As you mention, the bold green in the theoretical guarantees is indeed hardly noticeable, we have made it brighter. It was also hard to see because the differences in guarantees between meta-tailoring and classic (inductive) learning are very small (text-wise, not result-wise!), affecting only the subscript of a subscript. In particular, $f_{\theta_{x,S}}$ turns to $f_{\theta_{S}}$ for inductive learning guarantees, because meta-tailoring adapts the parameters to the input $x$, but regular inductive learning doesn’t.
> By having two extremely similar bounds, we can understand the effect of meta-tailoring more clearly.
>
> **Eliminating generalization gap for tailoring loss** Concretely, we show that we can upper bound the task loss using two terms. The first term is the key to our approach, depending solely on the tailoring loss at the query point. Because we tailor a custom $\theta_x$ to minimize the tailoring loss at test time, as well as training time, we can make the term small, eliminating the generalization gap that arises in classic ML training. In contrast, classic (inductive) ML cannot adapt $\theta$ to each $x$ and thus has a generalization gap. The second term involves a classic stability bound, and we also provide arguments why meta-tailoring can help on that term as well.
> In the general response to all reviewers, we address the confusion regarding the contrastive experiments and add more details on the theoretical guarantees. There, we also discuss new results on model-based RL.
>
> **Empirial validation**: please see the general response where we detail new baselines with related work and a totally new application in model-based RL, added on a new section 5.3 in the main text.

---

### Official Review · AnonReviewer1 · 2020-10-28
**Interesting idea, but the experiments are weak**

**Rating:** 4
**Confidence:** 3

**Review:**

This paper presents tailoring and meta-tailoring to eliminate the generalization gap by optimizing at test time. This paper combines the idea of test time optimization and meta-learning and provides some theoretical analysis of the proposed methods. Experiments show the proposed method is effective in solving the machine learning problem and improving model robustness.

Strengths:
- The idea of using meta-learning to improve tailoring is interesting. Some theoretical justification for the proposed method is provided.
- The explanation of different learning settings is insightful.

Weaknesses:
- The proposed tailoring method is very close to Test Time Training (Sun et al. , 2019). However, this paper fails to clearly show the differences between tailoring and TTT. There is also no theoretical or empirical evidence to show that tailoring/meta-tailoring is better than TTT.
- The experiments in this paper are quite weak. The descriptions of the experiment settings are unclear. There is no direct comparison with closely related methods like TTT. The results of widely used benchmarks are not provided.

Overall, although I think the idea is interesting, the proposed method is not well verified, which makes the effectiveness of the method is still unclear. Some closely related work is not sufficiently discussed. Therefore, I lean to recommend rejection for this paper.

---

> ### Author Response · Authors · 2020-11-24
> **Extra experiments and standard benchmarks on adversarials**
>
> Thank you for your comments.
>
> **Standard benchmarks on adversarial examples and model-based RL**
>
> "The results of widely used benchmarks are not provided"
>
> As you point out, the physics experiment is not a standard benchmark. We chose it for instructive purposes and to show potential useful applications of our method in science. However, the adversarial experiments build on arguably the most standard certifiable defense (Randomized Smoothing) and evaluate on the two most important benchmarks for adversarial examples: CIFAR-10 & Imagenet. For conciseness (and because we believe the value of the paper lies in its novelty and general applicability rather than matching state-of-the-art) we put the tables including the SOA methods in the appendix. We also mentioned both works in the penultimate paragraph of section 5.2 (Zhai et al., Salman et al.). There, we detail that our approach matches SOA on ImageNet (not CIFAR-10) despite our framework not being specialized to adversarial examples and that meta-tailoring could also potentially improve the other approaches.
>
> In the general response we detail new experiments improving on PDDM (Nagabandi et al.), a popular method, on a complex MuJoCo benchmark, with results added to section 5.3 in the main text. We believe these further show the wide range of applicability of meta-tailoring.
>
> **Extra experiments comparing to TTT**
>
> "There is no direct comparison with closely related methods like TTT."
>
> In the general answer to all reviewers we discuss differences from test time training (Sun et al.), and show that it is a sub-optimal approach for same-distribution generalization by providing extra experiments. We hope this addresses your concerns, both in terms of novelty and experimental performance.

---

### Official Review · AnonReviewer3 · 2020-10-29
**Very good paper with strong theoretical analysis**

**Rating:** 6
**Confidence:** 3

**Review:**

==========
Summary

This paper proposes a meta-tailoring method where the auxiliary tailor loss is used to adapt the model parameters at test time.  The paper also provides a theoretical analysis of the advantages of the proposed tailoring/meta-tailoring.  The proposed method is evaluated on various application tasks and obtain significant improvements over baselines.

==========
Pros

1. Test-time training is getting increasing attention recently, however, the theoretical analysis of the advantages of test time training is still behind. This paper provides a reasonable theoretical analysis of tailoring and meta-tailoring.
2. The proposed CNGrad approach is simple but effective, which provides strong empirical improvements on various tasks.
3. The paper is well written and clearly organized.

===========
Concerns/confusions

1.  what is the formulation of the affine parameters?  Is there any study about adapting the entire model versus the affine parameters only?

2. Missing contrastive learning experiments on images?  In the last sentence of the abstract, it says "..., and using contrastive losses on the query image to improve generalization". However, I didn't find a discussion on it in the main paper.  Given that previous work (e.g., Sun et al.) has done experiments on image classification tasks, it would be more compelling to provide a comparison in the same setting.


======

In general, this paper proposes an interesting approach and insightful analysis of the proposed method.  I'm willing to upgrade my score with comparable experiments with prior work on the image classification task.

---

> ### Author Response · Authors · 2020-11-24
> **clarifications on affine parameters + pointing to general answer**
>
> Thanks for your review.
>
> **“What is the formulation of the affine parameters? Is there any study about adapting the entire model versus the affine parameters?”**
>
> - Inspired by conditional normalization, our method CNGrad adds affine layers $\vec{y} = \vec{\gamma}\cdot \vec{x} + \vec{\beta}$ and only optimizes these parameters in the inner loop. These layers can be added in every hidden layer of most deep architectures.
> - The closest method that we know is CAVIA (Zintgraf et al.), which makes a neural network predict these affine parameters from a trainable vector. Then, instead of directly optimizing the affine parameters, they optimize the trainable vector. We show this extra complexity is unnecessary.
> - In theorem 2 of section 4 we prove that optimizing the affine parameters alone has the same capacity as optimizing the entire network. This theorem holds for a wide variety of losses and realistic network requirements, but requires evaluating the loss only at a finite number of data-points. As a result, the affine parameters have enough capacity for the inner loop (which uses a finite number of samples), but not for the outer loop (where we optimize all parameters to express arbitrary functions).
>
> **Comparison to test-time training, missing contrastive results and new model-based RL results:**
>
> We address these points in the general answer, where we also explain a new application of meta-tailoring to model-based RL, added in section 5.3 in the main text.

---

### Author Response · Authors · 2020-11-24
**General response: New experiments on RL application + test-time training baseline with discussion on differences**

Thanks for your reviews. Here we go over common comments and explain new results:

**Differences w.r.t. prior work and extra experiments to elucidate these differences**

Fine-tuning has long been a popular idea to adapt to new distributions or new tasks. To adapt to new distributions or tasks at test time, when supervision is not available, multiple methods have proposed to fine-tune using unsupervised objectives. In particular, in “Test time training(TTT) with self-supervised objectives for out-of-distribution(OOD) generalization”, Sun et al. address the case of distribution shift between training and testing. Tailoring also optimizes an unsupervised objective, but on a different problem and for a different reason. We study the standard ML setting in which test and training share both task and distribution. There, one would think fine-tuning is not necessary because there is nothing to adapt. However, inspired by transductive learning, we show that it is useful to tailor our models to each input to encode inductive biases at prediction-time.

These conceptual differences have three practical consequences:

1. **Whereas TTT only makes sense at test-time, when there is a different distribution, tailoring also makes sense at training time: meta-tailoring.** Moreover, this work shows theoretically and empirically that meta-tailoring is a better paradigm than tailoring. This is because, in meta-tailoring, training and test use the same predictive mechanism, which includes the tailoring step.

2. **In contrast to prior work, which cares about adaptation, tailoring improves the standard ML setting; being widely applicable.** In experiments explained in point 3, we show TTT is not good in this same-distribution setting. Moreover, the theoretical guarantees of meta-tailoring are much stronger than TTT’s. In particular,TTT has to make a posteriori assumptions about the gradients of the model once trained (which may not be true in the actual model). In contrast, ours are a priori: we ensure training will go in such a way that guarantees will be met. Note that, conversely, meta-tailoring is probably not the right approach for the out-of-distribution generalization, the setting of the TTT paper(see app. F).

3. TTT proposes to adapt the model by optimizing unsupervised objectives on available samples from the test distribution. Tailoring instead proposes to customize the model to each individual input. **We made two experiments comparing both hypothesis:**
    - In TTT, we care about adapting to the test distribution with any available samples. Therefore, the sample we use to adapt should be mostly irrelevant. However, we find this is not the case in the standard ML setting: **using the same sample $x$ to adapt and evaluate(tailoring) is significantly better than adapting on another test sample** $x'\neq x$ before evaluating on $x$ (2.9% vs 7.6% improvement).
    - In TTT, given more test samples, performance should increase and outperform tailoring. We show this is not the case when the distributions are the same: **even adapting with 6400 samples (a full batch, including the evaluated sample), performance of TTT is worse than tailoring(3.5% vs 7.6%  improvement)**. Meta-tailoring (36%) further increases this gap. We have updated table 1 and figure 1 to include it, although note that it is not a valid approach because test samples should be evaluated independently.

**Lack of contrastive learning results:**
Some reviewers pointed out the original text wasn’t clear about the lack of contrastive experiments, we are sorry for the confusion. We used the word “applications” when we should have used “domains” and over-compressed some sentences; we have now improved the wording. Making real experiments out of the theory has two big hurdles:
- We carefully chose the contrastive loss to make the theory as clean and insightful as possible. In particular, the meta-tailoring bound is the same as the inductive bound just changing $\theta_{S}$ to $\theta_{x,S}$, easing comparisons. However, the crafted contrastive loss is hard to implement and computationally inefficient.
- Contrastive semi-supervised learning is dominated by methods requiring big servers (MoCo, SimCLR) just to train one model, making experiments very expensive.

**Addition of model-based RL results to the main text (section 5.3):**
We show the applicability of meta-tailoring to model based reinforcement learning (MBRL). We improve PDDM (Nagabandi et al.), a popular MBRL method. We reimplemented it from tensorflow to pytorch and tried the two complex domains from that paper. We only report one domain (dclaw), because we were unable to reproduce the results in the other domain using the base PDDM (both the original tensorflow and ours). We have been working with the PDDM authors to reproduce the other domain in order to add extra results in case of acceptance.

We believe these MBRL results further broaden the range of applicability of the tailoring paradigm.

---

### Decision · Program_Chairs · 2021-01-07
**Final Decision**

**Decision:**

Reject

**Comment:**

This work proposes new learning algorithms that fine-tune ("tailor") a model at test-time using unsupervised objectives. This formulation allows for introducing an inductive bias into the model that might improve generalization on unseen data. The proposed algorithm is demonstrated on two example tasks.

The reviewers like the topic and also find the proposed approach to be interesting. However, they are unconvinced by the current empirical evaluation of the method. Additional experimental evaluation could improve our understanding of the proposed method and help contrast it to previously proposed techniques. Given these reviews I recommend rejecting the paper at this time.